



# Intercomparison and Evaluation of Ground- and Satellite-Based Stratospheric Ozone and Temperature profiles above Observatoire Haute Provence during the Lidar Validation NDACC Experiment (LAVANDE)

Robin Wing[1], Wolfgang Steinbrecht[2], Sophie Godin-Beekmann[1], Thomas J. McGee[3], John T. Sullivan[3], Grant Sumnicht[3], Gerard Ancellet[1], Alain Hauchecorne[1], Sergey Khaykin[1], and Philippe Keckhut[1]

[1]LATMOS/IPSL, OVSQ, Sorbonne Universités, CNRS, Paris, France
[2]Deutscher Wetterdienst, Met. Obs. Hohenpeissenberg, Hohenpeissenberg, Germany
[3]NASA Goddard Space Flight Center, Greenbelt, Maryland

**Correspondence:** Robin Wing (robin.wing@latmos.ipsl.fr)

**Abstract.** A two-part inter-comparison campaign was conducted at L'Observatoire de Haute Provence (OHP) for the validation of lidar ozone and temperature profiles using the mobile NASA Stratospheric Ozone Lidar (NASA STROZ), satellite overpasses from the Microwave Limb Sounder (MLS), the Sounding of the Atmosphere using Broadband Emission Radiometry (SABER), meteorological radiosondes launched from Nîmes, and locally launched ozonesondes. All the data were submitted

and compared "blind", before the group could see results from the other instruments. There was good agreement between all ozone measurements between 20 and 40 km with differences of generally less than 5% throughout this region. Below 20 km SABER and MLS measured significantly more ozone than the lidars or ozone sondes. Temperatures for all lidars were in good agreement between 30 and 60 km with differences on the order of ±1 to 3 K. Below 30 km, the OHP lidar operating at 532 nm has a significant cool bias due to contamination by aerosols. Systematic, altitude varying bias up to ±5 K compared to

the lidars was found for MLS at many altitudes. SABER temperature profiles are generally closer to the lidar profiles, with up 3 K negative bias near 50 km. Uncertainty estimates for ozone and temperature appear to be realistic for nearly all systems. However, it does seem that the very low estimated uncertainties of lidars between 30 and 50km, between 0.1 and 1 K, are not achieved during LidAr VAlidation NDacc Experiment (LAVANDE). These estimates might have to be increased to 1 to 2 K.

## 1 Introduction

The international Network for the Detection of Atmospheric Composition Change (NDACC, http://www.ndacc.org), formerly the Network for the Detection of Stratospheric Change (NDSC), is composed of more than 70 research stations worldwide (Kurylo et al., 2016; De Mazière et al., 2018). Ground-based remote sensing techniques measuring atmospheric parameters such as temperature and trace gas concentrations are used in NDACC to allow 1) early detection of long-term changes in the atmosphere; 2) validation of atmospheric measurements from satellites; 3) investigation of connections between atmospheric





composition and climate change; 4) and to provide support for testing and improving numerical computer models of the atmosphere.

Ground based NDACC lidar stations have been providing routine long-term vertical profiles of stratospheric ozone and temperature since the mid-1980s (Steinbrecht et al., 2009a). One key lidar station is the Observatoire de Haute-Provence (OHP)
in Southern France, situated at 43.94° N, 5.71° E, and 650 m above sea level (http://www.obs-hp.fr/geo/geo_ohp.shtml). The first stratospheric ozone measurements at OHP started in 1977 (Megie et al., 1977), with routine measurements since 1985 (Godin et al., 1989). Dedicated temperature lidars at OHP have been providing routine stratospheric and mesospheric temperature profiles since 1978 (Hauchecorne and Chanin, 1980). A lidar for tropospheric ozone has been operating routinely since 1990 (Ancellet and Beekmann, 1997).

NDACC requires standardised, consistent, high quality, long-term measurements. Regular instrument and algorithm inter-comparison campaigns are used to validate NDACC instruments and to track possible instrument biases. NDACC lidars, for example, have been intercompared in the 1989 Stratospheric Ozone Intercomparison Campaign at Table Mountain, California (STOIC, Margitan et al., 1995); the 1995 Ozone Profiler Assessment at Lauder, New Zealand (OPAL, McDermid et al., 1998); the 1997 OTOIC intercomparison at Haute-Provence (Braathen et al., 2004); the 1998 Ny-Ålesund Ozone Measurements Inter-
comparison on Spitzbergen, Norway (NAOMI, Steinbrecht et al., 1999); the 1999 DIAL algorithm intercomparison campaign (Godin et al., 1999); the 2005 Hohenpeissenberg Ozone Profiling Experiment in Germany (HOPE, Steinbrecht et al., 2009b); and the 2009 Measurements of Humidity in the Atmosphere and Validation Experiments at Table Mountain, California (MO-HAVE, Leblanc et al., 2011). Many of these campaigns have resulted in corrections and improvements for the involved lidar systems and their analysis software. A review of NDACC validation exercises was done by Keckhut et al. (2004). In general,
the intercomparisons have shown that NDACC lidars can measure the stratospheric ozone profile with an accuracy better than 3% between 12 and 35 km altitude and better than 10 % between 35 and 40 km. For temperature, NDACC lidars are typically precise to better than 1 K from 30 to 40 km altitude, with precision decreasing above to e.g. 5 K near 70 km depending on the particular lidar station and integration time. These campaign findings are consistent with recent re-evaluations of theoretical uncertainty budgets by (Leblanc et al., 2016a, b, c).

In addition to the NDACC campaigns which primarily focus on stratospheric ozone, there have been a few recent NDACC-like lidar inter-comparisons for tropospheric ozone in the Tropospheric Ozone Lidar Network (TOLNet). The 2014 series of campaigns at five sites in the United States and Canada (DISCOVER-AQ and FRAPPÉ, Wang et al., 2017); the 2015 LaRC Ozone Lidar intercomparison in Hampton, Virginia (LaRC, Sullivan et al., 2015); and the 2016 Southern California Ozone Observation Project (SCOOP, Leblanc et al., 2018). Tropospheric ozone concentrations from the ozonesondes regularly
launched at OHP have been also frequently compared to the tropospheric ozone lidar data operated at the same site (Beekmann et al., 1995; Gaudel et al., 2015).

The purpose of the present paper is to report on the LidAr VAlidation NDacc Experiment (LAVANDE), which took place in July 2017 and March 2018 at the Observatoire de Haute-Provence (OHP) in Southern France. LAVANDE allows the comparison of the measured ozone profiles from the stationary differential absorption lidars for stratospheric (LiO$_3$S) and tropospheric
ozone (LiO$_3$T) at OHP (Godin-Beekmann et al., 2003; Ancellet and Beekmann, 1997) with ozone profiles measured from the



mobile trailer-based NDACC stratospheric ozone reference lidar (NASA-STROZ), operated by NASA's Goddard Space Flight Center (McGee et al., 1991). Additional comparisons are made with routine Electro-Chemical-Cell (ECC) ozone sondes flown at OHP, and with satellite measurements by the Microwave Limb Sounder (MLS, Waters et al., 2006) and the Sounding of the Atmosphere using Broadband Emission Radiometry instrument (SABER, Russell III et al., 1999). Except for the $LiO_3T$, all

these instruments also provide temperature profiles over a substantial part of the stratosphere. The lidar temperature profiles taken during LAVANDE are derived from the non-absorbing 355 nm line of the two ozone lidars ($LiO_3S$ and NASA-STROZ) and from the dedicated stratospheric and mesospheric temperature Rayleigh lidar at OHP (Hauchecorne and Chanin, 1980), nowadays using a Nd:YAG laser at 532 nm. These temperature profiles are compared with the routine radiosondes from the nearby Meteo-France station at Nîmes (43.86° N, 4.41° E, about 100 km west of the OHP station), and with routine strato-

spheric meteorological analyses from the US National Center for Environmental Prediction (NCEP).

It is important to note that LAVANDE was a "blind" intercomparison. All the data were collected by an impartial referee (W. Steinbrecht), who was not involved in running the campaign. Data from each ground-based instrument were submitted "blind" to the referee, within days (or maximum weeks) after the measurement, and without seeing results from the other instruments. The referee also carried out all the comparison data analysis.

## 2   Instruments used for LAVANDE

Table 1 summarises all the different systems participating in the LAVANDE intercomparison. Ozone profiles taken by the Stratospheric Aerosol and Gases Experiment III (SAGE-III) satellite instrument onboard the International Space Station (ISS) (Mauldin III et al., 1998) in solar or lunar occultation geometry were also considered for the LAVANDE intercomparison. However, the number of reasonably coincident SAGE-III profiles turned out to be too low for statistically meaningful results

(only 3 or 4 profiles). Therefore SAGE-III ISS profiles are not included here.

In addition to Table 1, each instrument in the intercomparison campaign is described briefly below. Key aspects are noted in each subsection. References to original or most recent instrument descriptions are given for those seeking further details.

### 2.0.1   OHP Stratospheric Lidar ($LiO_3S$)

The Stratospheric Ozone Lidar ($LiO_3S$) is a differential absorption lidar which relies on the difference in the absorption

cross-section for ozone at two different wavelengths. The DIAL technique infers the ozone number density by taking the derivative of the ratio between a strongly absorbed line (on-line) and a weakly absorbed line (off-line) (Pelon et al., 1986). The system at OHP has two lasers emitting in the ultraviolet at 308 nm (on-line) and at 355 nm (off-line), a constellation of 4 receiver telescopes, and a Horiba Jobin Yvon holographic grating for line selection, described in Godin-Beekmann et al. (2003). In addition to making measurements of ozone, the off-line of a DIAL system (355 nm) can be used to calculate

Rayleigh temperature (Hauchecorne and Chanin, 1980). The LAVANDE campaign represents the first attempt to validate $LiO_3S$ temperature profiles within the framework of NDACC. The comparisons made during this campaign will prove vital for the assessment of the temperature uncertainty budget. Measurements with this instrument have been ongoing since 1985





and to date amount to 3,678 nights of data. Further details can be found for ozone profile retrieval, error analysis, and vertical resolution determination in Godin-Beekmann et al. (2003) and for temperature profile retrieval in Wing et al. (2018).

### 2.0.2 OHP Tropospheric Lidar (LiO$_3$T)

The Tropospheric Ozone Lidar (LiO$_3$T) is also a DIAL system, however, it differs from it's stratospheric counterpart in a few key ways. The tropospheric DIAL system doesn't rely on two separate lasers to generate the on-line and off-line wavelengths. The laser source is a Nd:YAG laser fourth harmonic emission at 266 nm. Two additional wavelengths are generated at 289 nm and 316 nm through a process known as Stimulated Raman Scattering in a high pressure deuterium cell. Further details of this technique can be found in (Papayannis et al., 1990; Milton et al., 1998). Both photocounting and analog detection are applied to provide vertical profiles in the altitude range 2.5-15 km (Ancellet and Beekmann, 1997). The tropospheric ozone lidar has made continuous twice-weekly measurements since 1990 (Gaudel et al., 2015).

### 2.0.3 OHP Temperature and Aerosol Lidar (LTA)

The Lidar Température et Aérosols (LTA) is a classic Rayleigh–Mie–Raman lidar operating at 532 nm (Keckhut et al., 1993). The absolute temperature profile is directly derived from the range-square corrected lidar return signal (Hauchecorne and Chanin, 1980). The system employs a high powered laser transmitter and a constellation of 4 receiver telescopes. It has been making regular measurements since 1978. Further details about this instrument and the most recent technical specifications can be found in (Wing et al., 2018).

### 2.0.4 NASA Stratospheric Ozone Lidar (NASA STROZ)

NASA's Goddard Space Flight Center Stratospheric Ozone Lidar (NASA STROZ) is a mobile validation lidar which is shipped across the world on a regular basis to run intercomparison and validation campaigns with ozone and temperature lidars in NDACC. The NASA STROZ is a DIAL system similar to the LiO$_3$S, relying on an on-line wavelength of 308 nm and an off-line wavelength of 355 nm. The system was orginally constructed in 1988 (McGee et al., 1991) and has been used as a reference during campaigns for multiple lidar stations since then (McGee et al., 1995).

### 2.0.5 Radiosondes and Ozonesondes (ECC)

Electrochemical Concentration Cell ozonesondes (ECC) manufactured by ENSCI-Z filled with 1% of potassium iodide (KI) and coupled to MeteoModem M10 radiosondes were launched every two nights during the first phase of the campaign in July 2017, and nightly during the second phase of the campaign in March 2018. The campaign ECCs reached a median burst altitude of 32.7 km with only one balloon bursting early at 17 km. Below 21 km, in the first phase of the campaign, the sondes flew north at the beginning of July, west near mid-month, and south by the end of the month. Above 21 km, all the 2017 sondes were carried east by the prevailing summer stratospheric wind. During the second phase of the campaign, the sondes flew generally north with only slight westerly changes in trajectory as they ascended. ECC ozonesondes provide a precision of ±(3–5)%, and





an accuracy of ±(5–10)% (Smit, 2013; Tarasick et al., 2016). A known positive bias of the ENSCI ECC data in the troposphere when using 1% KI concentration (Smit et al., 2007), is corrected by decreasing the ECC ozone concentration by 4% below the tropopause (Gaudel et al., 2015). Weekly ECC launches have been conducted at OHP since 1991 and a correction factor (fc) is calculated using a normalisation of the total ozone from the sonde to the total ozone measured by a SAOZ spectrophotometer

at OHP (Nair et al., 2012; Gaudel et al., 2015). The ECC data is discarded if the calculated correction factor, fc, is outside the range of 0.8-1.2. The correction factor is not applied to the ECC data and measured ozone partial pressure are not corrected above the tropopause. During the LAVANDE campaign, the correction factor is always in the range 0.92-1.05 except for on March 20 when fc=1.16.

In addition to the ECCs, we also used the MeteoModem M10 meteorological radiosondes launched twice daily from the

nearby station at Nîmes.

## 2.1 Co-located satellite overpasses

The satellite based MLS (Microwave Limb Sounder) and SABER (Sounding of the Atmosphere using Broadband Emission Radiometry) instruments provide stratospheric ozone and temperature profiles over most of the globe.

### 2.1.1 Microwave Limb Sounder (MLS)

The Microwave Limb Sounder (MLS) is a spectrometer aboard the Aura satellite which measures thermal microwave radiation from the atmosphere in limb geometry and allows retrieval of stratospheric ozone profiles with a vertical resolution of about 3 km and retrieval of stratospheric temperature profiles with a typical vertical resolution of 8 km at 30 km altitude, 9 km at 45 km altitude, and 14 km at 80 km (full width at half maximum (FWHM) of the averaging kernels, Schwartz et al., 2008). We have used version 4.0 MLS profiles of temperature, geopotential height and ozone. A more complete description of the instrument

is given in Waters et al. (2006).

### 2.1.2 Sounding of the Atmosphere using Broadband Emission Radiometry (SABER)

The Sounding of the Atmosphere using Broadband Emission Radiometry (SABER) instrument onboard the Thermosphere Ionosphere Mesosphere Energetics and Dynamics (TIMED) satellite makes ozone and temperature measurements from about 15 to 100 km. For temperature, it provides a vertical resolution of 2 km and temperature accuracy of 1 to 2 K between 15

and 60 km, decreasing to 5 K near 85 km, and to 10 K near 100 km (Rezac et al., 2015a, b). For ozone, SABER provides 1% precision between 40 and 50 km altitude, decreasing to 2% near 30 and 55 km and to 10% near 15 and 80 km (Rong et al., 2009). We have used version 2.0 SABER profiles of temperature and ozone. A more complete description of the instrument is given in Mertens et al. (2001).





### 2.1.3 Co-locating satellite profiles and ground-based profiles.

While all the lidars were measuring at the same location and the same time during LAVANDE, and the ECC sondes were quite close in time and space, satellite profiles almost never match the exact time and location of a ground-based measurement. For LAVANDE, we considered all satellite profiles with a tangent point within $\pm 5°$ latitude and $\pm 15°$ longitude of the OHP station

(43.94° N, 5.71° E), and within $\pm 12$ hours of 00 UTC (1 hour after local midnight for the lidar measurements nights) (see also Wing et al., 2018b). This fairly large coincidence box is depicted in Fig. 1. It covers most of Southern Europe, from Paris in the North to the southern tips of Spain or Sardinia in the South, and from Portugal in the West to Slovakia, Hungary, or Serbia in the East. The size of the chosen box size is a matter of compromise. On the one end, a small coincidence box results in very few coinciding satellite profiles, but also very close matches in time and space between satellite and ground-based profiles. On

the other end, a large box results in many coinciding satellite profiles, but poor matches in space and time. The box size chosen here is similar to the compromise chosen in Wing et al. (2018b). It results in 10 to 20 matching MLS and SABER profiles for all lidar nights during the LAVANDE campaign.

The question then arises, which of these 10 to 20 profiles should be used for the intercomparison. One choice would be to take the profile that matches most closely in space and time. Another choice would be to use the average profile obtained from

all satellite profiles in the coincidence box. A third possibility is to use the weighted average profile, with lower weight given to satellite profiles that are further away in space or time. We used weights proportional to one over the $\sqrt{(\Delta r^2 + (v \cdot \Delta t)^2)}$, where $\Delta r$ and $\Delta t$ are the distance in space and time between the lidar profile and the satellite profile, and $v = 10 m/s$ is a wind speed typical for the mid-stratosphere. For the LAVANDE intercomparison we tested these three possible profile choices. Generally, differences between all three choices were quite small. Overall, however, the weighted average profile gave slightly

better results than the others. Therefore, the weighted average MLS and SABER profiles are used throughout most of this paper.





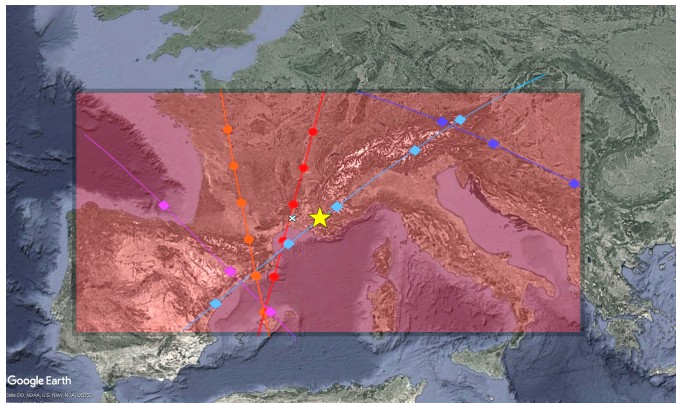

**Figure 1.** The area defined for coincident measurements during the LAVANDE campaign (39,-9) to (49,21). L'Observatoire de Haute Provence is represented by the yellow star at (43.93,5.71) and Nîmes radiosonde launches by a cyan X at (43.86,4.41). Ascending (red) and descending (orange) orbits for MLS with tangent point locations of profiles for 17 July 2018. Ascending (light blue) and descending (purple and dark blue) orbits for SABER with tangent point locations of profiles for 17 July 2018. (data: © Google Earth Pro, 2019)

## 3    Campaign Overview

The LAVANDE campaign took place in two parts: the first period covered about two weeks in summer 2017, from July $10^{th}$ to $26^{th}$ and the second period covered 10 days in early spring 2018, from March $12^{th}$ to $22^{nd}$. Table 2 shows which ground-based systems provided ozone and/or temperature profiles on each of the different nights of the campaign. Temperature profiles from NCEP reanalysis were included as well. Overall, LAVANDE covered about 4 weeks of measurements, and provided ≈ 120 ground-based temperature profiles, and ≈ 60 ground-based ozone profiles. Due to a laser failure in the NASA-STROZ system, that system was not able to measure ozone profiles after July $18^{th}$ in 2017. Temperature measurements however, were still possible and were not affected. The NASA-STROZ laser was repaired by March 2018 for the second phase of the campaign. All other systems were operating nominally throughout the campaign with no significant problems. The MLS and SABER satellite instruments provided ozone and temperature profiles during all campaign nights, in the spatial and temporal coincidence box introduced in Fig. 1.

### 3.1    Example Comparisons

Two examples for both ozone and temperature profiles for a LAVANDE night in July 2017 and March 2018 are given in Figs. 2 and 3. We can see the high degree of fidelity in reproducing the ozone profile across all ground based instruments. In particular, we see very good agreement of the small scale features present below 15 km in the July example. In Fig. 2 we see that the ozone number density is fairly low throughout the troposphere, about $1 \times 10^{12} cm^{-3}$, slightly declining up to the tropopause at about 13 to 15 km. Above the tropopause, ozone increases substantially up to the number density maximum, located at about 25 km altitude in July 2017 and about 19 km in March 2018. In the left hand panel, above the ozone maximum, ozone decreases





steadily with altitude, from about $4 \times 10^{12} cm^{-3}$ near 25 km to less than $1 \times 10^{12} cm^{-3}$ near 50 km. In the right hand panel, we see much more variation in the upper troposphere and lower stratosphere which is consistent with the more dynamically variable spring at OHP. Additionally, the March ozone maximum is greater and lower in altitude, about $7 \times 10^{12} cm^{-3}$ at 18 km. In general, the ozone profiles have less vertical structure and are smoother above 25 km. It is important to note that the lower

stratospheric ozone is much more variable in the spring time (left panel) than in the summer in response to seasonal dynamics. This increased variability introduces an added layer of complexity to our analysis and must be accounted for carefully.

In order to compare the ozone profiles from the different systems, it is necessary to put the data on a common altitude grid. For LAVANDE a vertical grid with 300 m spacing was chosen. Data with finer vertical spacing (lidars and sondes) were averaged to 300 m wide altitude bins centred around the mid-points of this grid. Data with coarser vertical spacing (satellites

and NCEP) were interpolated to the 300 m grid. In the troposphere and lower stratosphere up to about 25 km the conversion to the 300 m vertical grid smooths out some of the finer structures present in the original lidar data whereas, at higher altitudes the differences between the original data and the data on the 300 m grid are small. For most instruments, the lack of finer structures above 30 km is due to limited vertical resolution of the original retrieved profiles.

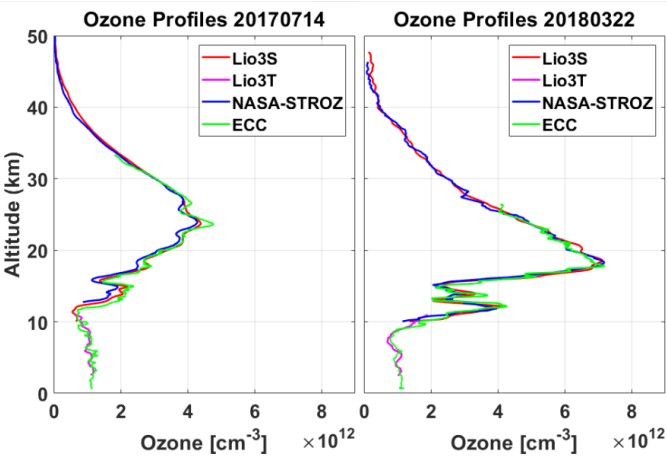

**Figure 2.** Ozone profiles measured by the different instruments at Observatoire de Haute-Provence on the nights July 14/15, 2017 and March 22/23, 2018. Note the seasonal differences in the character of the ozone profiles in spring and summer.

The temperature profiles in Fig. 3 are for the same night from July 2017 and March 2018 and show the usual temperature

decline throughout the troposphere. On the July night, the tropopause is located at about 13 km altitude and around 10 km in March. Above the tropopause, the temperature increases with altitude up to the stratopause at 45 to 50 km. There is a distinct difference in the temperature lapse rate of the lower stratosphere in the spring (right panel) as the atmosphere is nearly isothermal until 30 km. The increased spring time variance in the lower stratospheric temperatures should be considered when conducting lidar validation studies. In the mesosphere, from 50 to 80 or 90 km, temperatures decrease again with altitude.

Temperature profiles measured by all systems in Fig. 3 show these features with good consistency between systems over a wide altitude range. As with the ozone profiles in Fig. 2, conversion to the regular 300 m altitude grid smooths out finer structures at





lower altitudes. For temperature, the highest vertical resolution data, down to a few meters, come from the radiosondes coupled to the ECC ozone sensors. Lidar temperatures have vertical resolution of 150 m in the lower stratosphere to greater than 1 km in the mesosphere. The other systems have vertical resolutions which are generally coarser than 1 km.

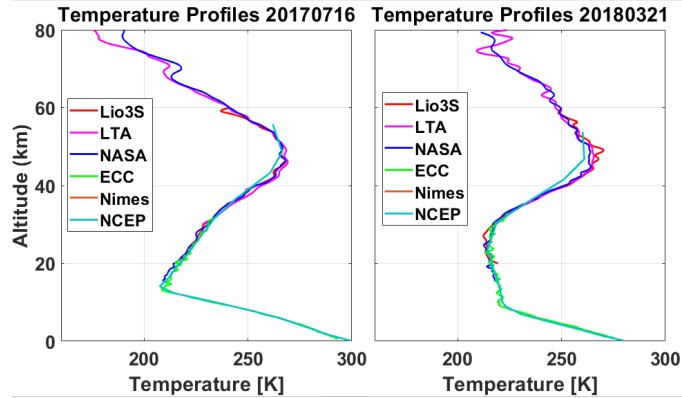

**Figure 3.** Temperature profiles measured by the different instruments at Observatoire de Haute-Provence on the nights July 14/15, 2017 and March 22/23, 2018. Note the seasonal differences in the character of the ozone profiles in spring and summer.

### 3.1.1 Comparisons with Satellites

Figures 4, ozone density, and 5, temperature, give examples from the second part of LAVANDE in March 2018 and also include MLS and SABER satellite data. There is generally good agreement between all instruments for both ozone and temperature profiles; all instruments show similar ozone profiles with the ozone maximum occurring near 20 km. The ground-based measurements also reproduce the fine scale ozone features as narrow as 150 m in vertical extent over a wide range of altitudes. All instruments correctly identify the tropopause and stratopause at same altitudes and amplitudes, to within 5 K. However,

the satellite profiles of both ozone and temperature can be vertically offset from the profiles produced by the ground based instruments. In Fig. 4 (left), we can see that the ozone maximum at 20 km and the sharp ozone decrease at 18 km, reported by the ground-based instruments, is shown 3 to 4 km lower in the MLS profile. This tendency for vertical offsets between lidar profiles and satellite profiles of temperature has been systematically documented over decades long time scales by Wing et al. (2018b) and is attributed to systematic errors introduced in the retrieval of geopotential height is the satellite profiles.

In the left panel of Fig. 4 we present a case with less than 10% difference (with the exception of MLS below 20 km) between ozone profiles measured by the lidars and the satellites. In the right panel is shown the percent difference for each profile with respect to the LiO$_3$S profile. We can see that MLS and LiO$_3$T agree fairly well between 5 and 11 km, following the same trend of ozone increasing with altitude. MLS is in relatively poor agreement with all other instruments between 11 and 20 km with negative biases reaching $-40\%$, as shown in the left panel, while the ozonesondes and lidars compare much better in

this region, with only 5% difference between them. The agreement between all measurements from 20 to 40 km is good, with percent differences less than 20%. Of particular interest is the region of disagreement between 11 and 20 km, characterised by rapid variation and spikes in the percent difference plot, where the low vertical resolution of MLS cannot resolve the fine



layers of the dynamic lower stratosphere. There is also a slight vertical offset in the altitude of the peak ozone concentration near 20 km in the satellite profiles.

In general, SABER ozone does not agree with ozone measurements from the other instruments below 25 km as it is principally an instrument focused on the upper middle atmosphere; hence it is not plotted for this altitude range. The extent of

the disagreement can be an order of magnitude larger than the differences between the ozone concentration measured by the other instruments. Presented in Fig. 4 is our best SABER comparison where we can see good agreement between SABER and the lidar above 25 km. SABER tends to report slightly higher ozone number densities above 30 km than other measurements. There is also a slight disagreement about the altitude of the peak ozone concentration and the overall thickness of the ozone layer.

One key point to keep in mind when interpreting the right panel of Fig. 4 is that in regions on either side of the ozone maximum, where ozone densities are low, the percentage differences can be quite large but only represent slight differences in the number density.

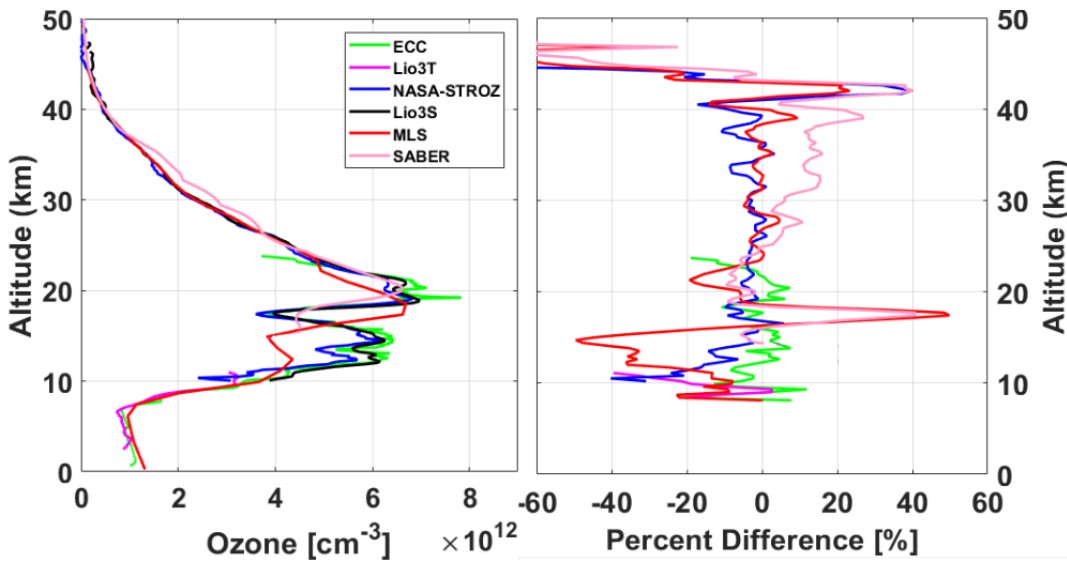

**Figure 4.** Satellite and lidar ozone profiles measured on the night March 19/20, 2018 at or near the Observatoire de Haute-Provence (left). Percent differences for each profile with respect to the LiO$_3$S profile (right). All profiles have been converted to the same 300 m vertical spacing altitude grid. For MLS and SABER, the weighted average profile is calculated based on the distance in time and space between the individual satellite profiles and the OHP station.

In Fig. 5, right panel, the temperature differences are plotted for each profile with respect to the NASA lidar temperature. We can see that all instruments agree fairly well with the NASA lidar up to 60 km with disagreements in the mesosphere. The

deviation of the LTA temperature profile from the NASA temperature profile below 30 km is a known cooling effect of the differential absorption of laser light by aerosols in the visible and UV. The 532 nm LTA system is more strongly influenced by stratospheric aerosols than the 355 nm NASA lidar and LiO$_3$S systems. There is a warm bias in LiO$_3$S below 20 km. As





the primary purpose of LiO$_3$S is the measurement of stratospheric ozone, the temperature retrievals, particularly those in the troposphere, are a value-added product of this system. The temperature measurements in the stratosphere compare very well with those of the other instruments, and with the addition of a new Raman channel, and a new comprehensive temperature retrieval package, it is anticipated that the warm bias evident below 20 km in Fig. 5 will be reduced.

Of particular interest is a small developing Mesospheric Inversion Layer present near 71 km which is seen by both the NASA and LTA lidars. MLS displays an evident kink in the temperature profile at 65 km which could be the signal of the inversion layer given that the satellite has an effective vertical resolution of nearly 15 km at those altitudes. SABER does not detect the layer on this night but does track the development of the feature over the next few nights.

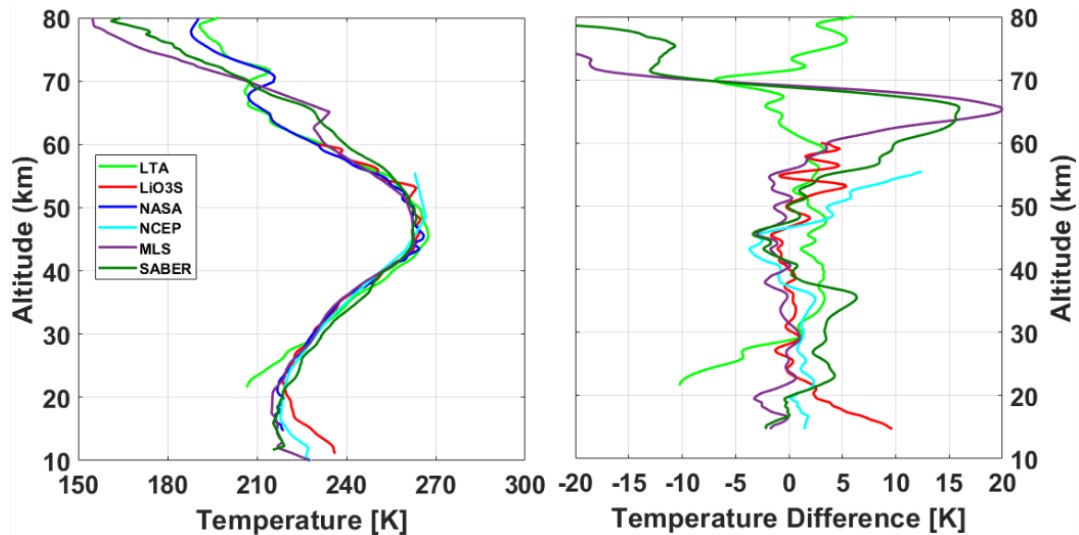

**Figure 5.** Satellite, NCEP, and lidar temperature profiles measured on the night of July 24/25, 2017 at or near the Observatoire de Haute-Provence (left) and temperature difference profiles with respect to the NASA temperature profile (right). All profiles have been converted to the same 300 m vertical spacing altitude grid. For MLS and SABER, the weighted average profile is calculated based on the distance in time and space between the individual satellite profiles and the OHP station.

## 4   Intercomparison Results for Ozone

Figure 6 shows the time series of ozone concentrations measured by the different systems for a number of selected levels. A clear separation can be seen between the two measurement periods in July 2017 and March 2018, due to the normal seasonal cycle. Ozone values in the lower stratosphere (below about 25 km) were higher in March 2018 than in July 2017. In the upper stratosphere (above 30 km), in contrast, ozone values were lower in March 2018. In addition, atmospheric conditions (and ozone values) were much more variable in March 2018. Generally, all instruments track ozone variations in a similar way.

However, Fig. 6 does indicate some systematic deviations. For instance, the NASA-STROZ lidar tends to report lower ozone





values near 40 km, while LiO₃S reports higher ozone concentrations than MLS, and SABER tends to report more ozone at lower levels.

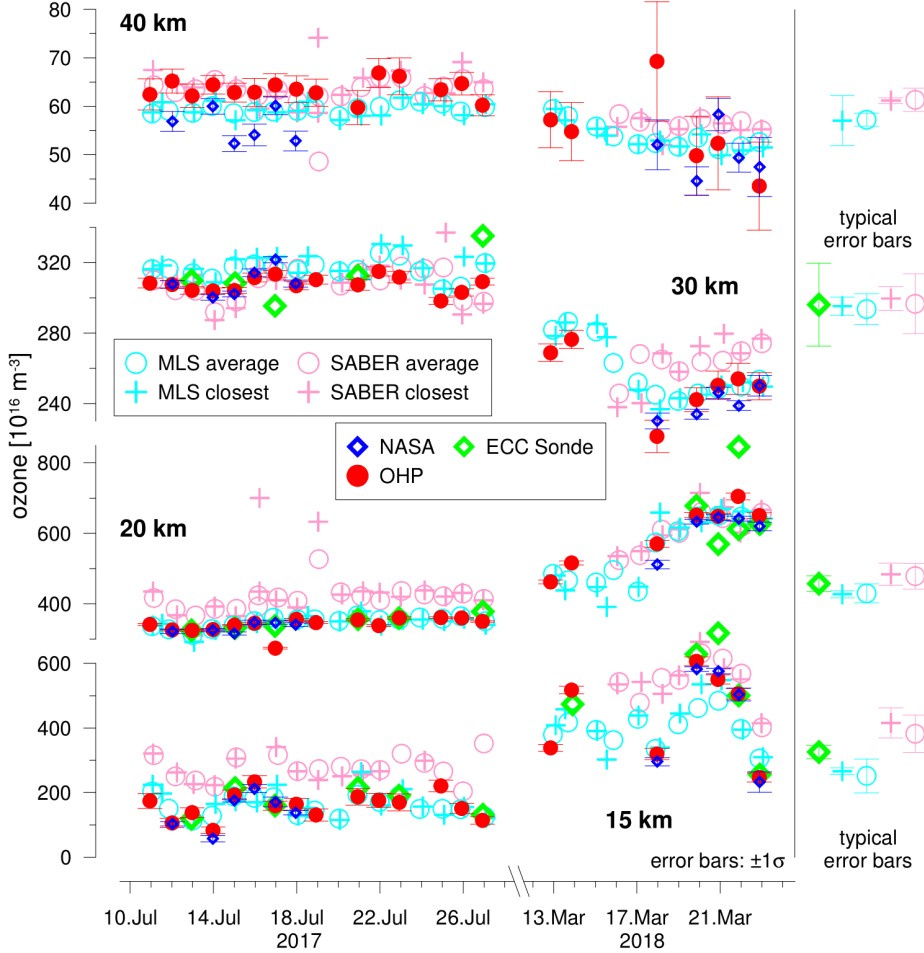

**Figure 6.** Time series of ozone concentrations measured at different altitude levels during LAVANDE.

A closer look at the systematic differences in the ozone profiles produced by each instrument, as well as their statistical uncertainty, is given in Fig. 7. This Figure shows the average relative difference profile between ozone from the various instruments and ozone from the LiO₃S. The LiO₃S was chosen here as a reference, because it had the most measurement nights of all ozone systems (due to the unfortunate laser failure of NASA-STROZ in July 2017). Similar to the results of previous NDACC intercomparisons (see introduction), the best agreement between the different ozone systems is found between 20 and 40 km altitude. During LAVANDE agreement over most of this altitude range was better than ±5% between most systems, with no statistically significant differences at 2σ (95% confidence level). SABER measured some larger and more significant differences up to ±10% at some altitudes. Above 30 km, the ECC sondes measured slightly lower ozone concentrations than the other instruments by up to −10%.



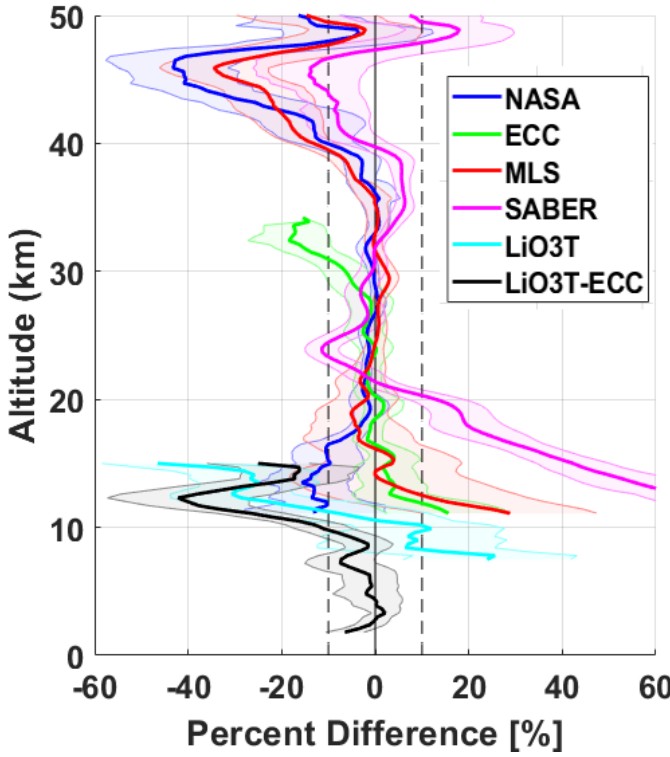

**Figure 7.** The average relative difference profile between the ozone profiles measured by the various LAVANDE instruments compared to the ozone profile measured by the LiO₃S. The shaded range gives the ±2 standard deviations of the mean and indicates the statistical confidence interval at the 95% uncertainty level. Results for MLS and SABER are reported using the weighted average profiles, but very similar results are obtained by using only the profile from the closest SABER or MLS overpass.

Below 20 km, and above 40 km, the ozone concentration profiles from the different systems show larger deviations. Around 45 km, for example, NASA-STROZ, MLS, and SABER give 40, 30, and 15% lower ozone values, respectively, than the LiO₃S system. These differences are statistically significant for at the $2\sigma$ level. Differences of this kind can be caused by the specific differential filter used at high altitudes above 40 km in the LiO₃S and NASA-STROZ retrieval software (see also Godin et al.,

5   1999). The heavier smoothing and integration is required above 40 km due to the drop in the lidar signal to noise ratio.

Below 20 km, SABER reports significantly higher ozone than the other systems. MLS also tends to report higher ozone, with differences up +20% near 12 km, compared to the LiO₃S. However, this is not statistically significant at the $2\sigma$ level. The ECC sondes tend to report up +5% higher ozone than the LiO₃S between 10 and 15 km, whereas NASA-STROZ tends to report less ozone, -12% on average near 10 km. These ECC and NASA-STROZ differences are also not statistically significant at $2\sigma$

10   above 15 km. Finally, Fig. 7 indicates that the LiO₃T was in good agreement with the ECC sondes and the OHP stratospheric DIAL below 9 km, when the ECC sondes are corrected by the 4% in the troposphere. This differences increase above 9 km to a maximum of -40% near 14 km. The large percent difference between LiO₃T and LiO₃S between 10 and 15 km is unsurprising



as both instruments are operating near their detection range limits (low signal to noise ratio and vertical averaging larger than 1 km for $LiO_3T$ and large sensitivity to systematic errors for the $LiO_3S$ near 10 km).

Another way of viewing the differences between the ozone profiles measured by the different instruments is to use scatter plots of ozone concentration as function of altitude (seen in Fig. 8). To plot the scatter between datasets we further integrated

the ozone profiles to 2 km resolution to reduce the high frequency components. The three panels show generally good tracking of ozone measured by each of the different instruments against ozone measured by the $LiO_3S$, over a substantial range of ozone concentration values. Some of the systematic differences appearing in Fig. 4 can further examined in the scatter plots. One prominent example is the sharp onset of a high ozone concentration bias in SABER data below 20 km with respect to the other instruments. Looking at the left panel of Fig. 8 which represents the ozone concentration in the UTLS (0 to 20 km) we can

see that the SABER (magenta) bias occurs most strongly at the lowest ozone concentrations. SABER profiles appear to have a lower ozone concentration limit of 2 to $3*10^{12}cm^{-3}$ and cannot match other instruments measuring below $2*10^{12}cm^{-3}$. We can also examine the behaviour of the MLS bias in Fig. 7 which abruptly changed from positive below 25 km to negative below 15 km. Again we can see in the left hand panel of Fig. 8 that the sharp change occurs at very low ozone concentrations. For concentrations above $1*10^{12}cm^{-3}$ MLS has a low bias with respect to all other instruments however, below $1*10^{12}cm^{-3}$ the

variance abruptly increases with the majority of points exhibiting a high bias. These satellite-lidar biases in measured ozone concentration are a convolution of an unknown real ozone bias, a bias arising from sampling different air, and a bias arising from the vertical resolution and smoothing of the satellites.

The central panel of 8 shows the scatter between ozone measurements in the region between 20 to 30 km (nominally near the altitude of the ozone maximum). We can see five tight clusters of data points which correspond to data points every 2 km. It

is important to note that the real differences in the ozone concentration at these altitudes is low, so we have a very low variance associated with each cluster of points. The right hand panel of Fig. 8 shows the tracking of ozone concentrations from 30 to 50 km and much like the central panel can be characterised by low variability and low variance. It is important to note that neither MLS nor SABER exhibit strong biases at these altitudes. Also, note that the comparison between the ECC and $LiO_3T$ (black) is only present in the left hand panel as the upper limit of the tropospheric lidar is around 12 to 15 km.



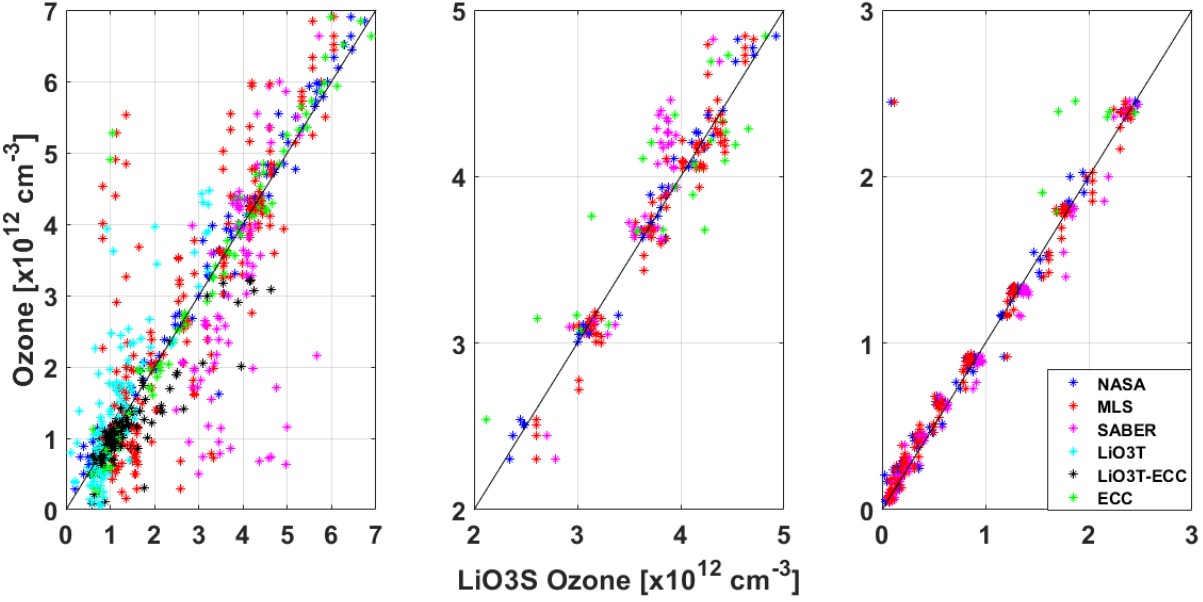

**Figure 8.** Scatter plots of ozone concentration as measured by the various LAVANDE instruments (along the vertical axis) and ozone measured by the LiO$_3$S (along the horizontal axis). Left: Ozone from 0 to 20 km altitude. Centre: Ozone from 20 to 30 km altitude. Right: Ozone from 30 to 50 km altitude.

A complementary method for tracking the 'goodness' of the match between the various LAVANDE instruments is presented in Fig. 9. It shows vertical profiles of the correlation between ozone from the each of the instruments and ozone from the LiO$_3$S. These correlations are taken using data from all LAVANDE nights (except outliers indicated in Tab. 2) which has been integrated to 2 km in an effort to filter out the high frequency components. Figure 9 shows very high correlation between ozone

concentration profiles measured by the LiO$_3$S and by NASA-STROZ (blue line) and between LiO$_3$S and the ECC below 20 km (green line). Over much of the 10 to 35 km altitude range, the correlations exceed 0.95 between the two stratospheric ozone lidars. A slightly surprising feature in Fig. 9 is the marked drop in correlation around 25 km near the maximum of the ozone concentration. This drop is due to the relatively low variability of real ozone in both time and altitude as was demonstrated in the central panel of Fig. 8. When the co-variance of the data, arising from real differences in ozone concentration drops faster

than the variance of the data, in part arising from statistical scatter, we see a resulting drop in the correlation. As a result, the drop occurs at altitudes where the combined sampling and instrumental uncertainty of each instrument play a larger role in the correlation than true variations in ozone. Rather unsurprisingly, this effect is most noticeable in the comparisons between the lidars and the satellites where the sampling and resolutions are most different. By varying the size of the window (number of data points / altitude range) used when calculating the correlations we can drastically increase or decrease the amplitude of this

peak. As such, the drop in the correlations at the ozone maximum should be considered as an artefact and not a true measure of geophysical differences. At other altitudes, ozone concentration varies much more over time and with altitude, giving more meaningful estimates of correlation.





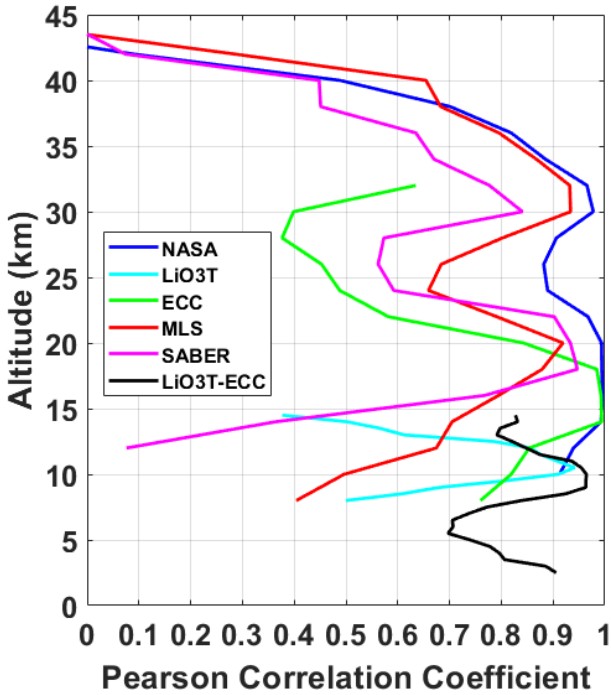

**Figure 9.** Vertical profiles of the correlation of ozone concentrations measured by the various LAVANDE instruments and ozone concentration measured by the LiO$_3$S (outliers were excluded). Correlation is taken over the 28 nights of the LAVANDE campaign and over 2 kilometres in altitude. Results for MLS and SABER are calculated from the weighted average profiles. Slightly smaller correlations were obtained for the closest match SABER or MLS profiles (not shown).

### 4.1 Ozone Uncertainty Analysis

Apart from the highlighted systematic differences and overall good tracking / correlation of the ozone concentration profiles, another important question we should ask is how realistic are the uncertainty estimates of the different systems? In the case of the lidars, the small number of photons scattered back from the stratosphere and detected by the lidar receiver on the

5    ground is generally the most important contributing factor to the measurement uncertainty (Godin-Beekmann et al., 2003; Leblanc et al., 2016a, b). Uncertainty sources for the ECC sondes include uncertain corrections for declining pump efficiency above 25 km, uncertain pressure / altitude registration, uncertain background current, evaporation of the sensing solution, and changing stochiometry in the chemical cell (Tarasick et al., 2016). The MLS and SABER satellite ozone retrievals also provide uncertainty estimates (Waters et al., 2006; Froidevaux et al., 2008; Rezac et al., 2015a, b).

10    As previously mentioned, additional complications arise due to substantial variations in real ozone concentration between the OHP lidar measurement and a SABER or MLS ozone profile which can be measured many hundred kilometres and several hours away. In principle, such real differences can also occur for the ECC sondes. However, the ECC sondes during LAVANDE were fairly close to the lidar profiles, particularly in the troposphere. They were launched at OHP during the time of the lidar



measurements and did not drift away by more than 100 km, even during the more variable weather and higher winds in the springtime part of the campaign.

Figure 10 shows the average relative ozone uncertainty estimated by the LiO$_3$S and the NASA-STROZ retrievals for nightly mean ozone profiles during LAVANDE. Both uncertainty profiles are comparable and have an uncertainty of less than 2%

between 20 and 35 km, with increasing uncertainty towards higher and lower altitudes. Below 15 km, the uncertainty is in the range 5% to 20% while above 35 km, the uncertainty increases to about 10% near 40 km, and to about 60% near 50 km. Very similar ozone uncertainties are reported in the comprehensive NDACC lidar uncertainty budget analysis of Godin-Beekmann et al. (2003) and Braathen et al. (2004). Assuming that there is no correlation between the average measurement noise of LiO$_3$S, $\sigma_L$ (red), and NASA-STROZ lidar, $\sigma_N$ (blue), in Fig. 10 then the relative standard deviation of the ozone difference,

$\sigma_{rel}$, between the two systems is given by Eq. 1 (black).

$$\sigma_{rel} = \frac{\overline{N}}{\overline{L}} \sqrt{\left(\frac{\sigma_L}{\overline{L}}\right)^2 + \left(\frac{\sigma_N}{\overline{N}}\right)^2} \tag{1}$$

If the uncertainty estimates are correct, it should be similar to the observed standard deviation of all the nightly mean ozone profile differences, $\sigma_{diff}$ (grey), expressed in Eq. 2 during LAVANDE.

$$\sigma_{diff} = \sqrt{\left(\frac{1}{N-1}\right) \Sigma \left(\left(\frac{N_i}{L_i}\right) - \left(\frac{\overline{N}}{\overline{L}}\right)\right)^2} \tag{2}$$

Apart from some additional noise (especially near 20 km), agreement between the relative standard deviation of the ozone difference and observed standard deviation of all the nightly mean ozone profile differences (black line and the grey line in Fig. 10) is quite good. From this agreement we have a strong indication that the ozone uncertainties provided by the LiO$_3$S and NASA-STROZ retrievals are realistic and we can proceed with our analysis.



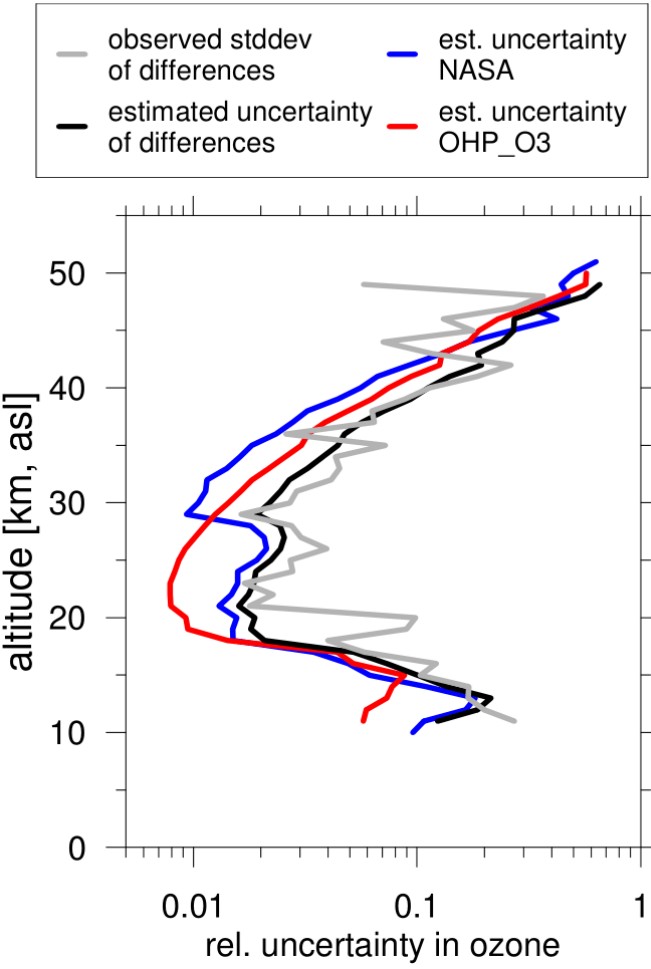

**Figure 10.** Vertical profiles of relative ozone uncertainties. Red: estimated by the LiO$_3$S retrieval. Blue: estimated by the NASA-STROZ retrieval. Black: estimated for the relative ozone difference between NASA-STROZ and LiO$_3$S ($O_3(NASA)/O_3(OHP)-1$). Grey: observed standard deviation for the relative ozone differences between NASA-STROZ and LiO$_3$S during LAVANDE.

Figure 11 shows similar results for the uncertainties of ECC sondes (green line) and the OHP stratospheric and tropospheric DIALs (red and orange lines). In this case, the estimated uncertainty of the relative ozone difference (black line) is dominated at most altitudes by the larger ozone uncertainty of the ECC sondes (green line). Again, agreement between estimated ozone difference uncertainty (black line) and the corresponding observed standard deviation (grey line) is quite reasonable. However, to achieve this level of agreement the estimate for ECC sonde ozone uncertainty from Tarasick et al. (2016) had to be doubled (to about 5% between 15 and 25 km, and to about 10% below 10 km and above 30 km). This would indicate that, at least during LAVANDE, the ozone concentration uncertainty for ECC sondes might be larger than estimated by Tarasick et al. (2016), see





also Smit (2013). It may improve once the homogenization of the OHP data set has been completed taking into account the use of 1% KI concentration in the stratosphere data processing ( 3-10%), and the humidification correction for the pump flow rate correction ( 1-4%) which are not currently applied.

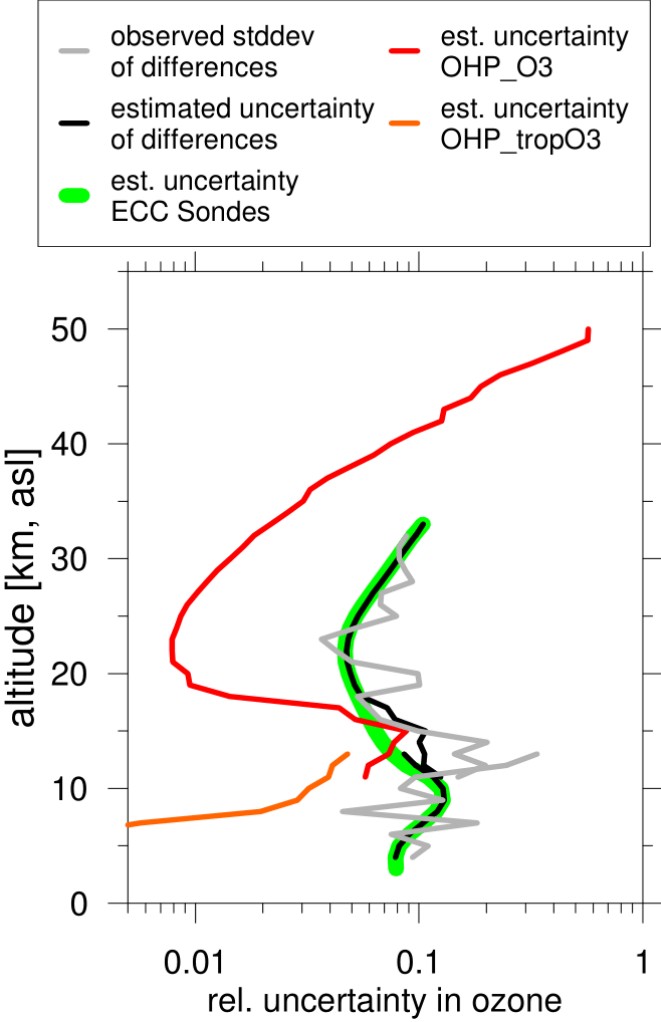

**Figure 11.** Vertical profiles of estimated relative ozone uncertainties for ECC sonde ozone profiles (green line, two times the estimate from Tarasick et al. (2016), excluding radiosonde pressure errors) and ozone uncertainty estimated by the $LiO_3S$ retrieval (red line), and the $LiO_3T$ (orange line). Black: estimated uncertainty for the relative ozone difference between ECC sondes and the two $LiO_3S$ (tropospheric system up to 13 km, stratospheric system above 10 km). Grey: corresponding observed standard deviation for the relative ozone differences during LAVANDE.

As previously discussed, the lidar measurements during LAVANDE were almost coincident in space and time, and the ECC sondes were very close. MLS and SABER satellite measurements, however, are usually taken several hours and several hundred





kilometres away from the lidar measurement. Therefore, substantial additional uncertainty in the relative ozone difference between MLS and the $LiO_3S$ arises from geophysical ozone variations. This "sampling uncertainty" can be estimated by the standard deviation of all satellite profiles in the previously discussed coincidence box (see Fig. 1). Note that this standard deviation includes both sampling uncertainty due to true ozone variation over the box and measurement noise of the individual

5  profiles.

The resulting uncertainties are shown in Fig. 12. At nearly all altitudes between 10 and 40 km, with the exception of 25 km where ozone variations are minimal (recall the discussion of the dip in the correlations in Fig. 9), the MLS sampling uncertainty (blue line) is clearly larger than the MLS individual profile uncertainty (cyan line). From 37 to 47 km, MLS sampling uncertainty and individual profile uncertainty are comparable, indicating that the estimate for individual profile uncertainty

10  is realistic and that geophysical ozone variability at these altitudes is small in comparison. However, above 47 km sampling uncertainty is actually smaller than the estimated individual profile uncertainty - indicating that the MLS profile uncertainty estimate may be too conservative in this region.





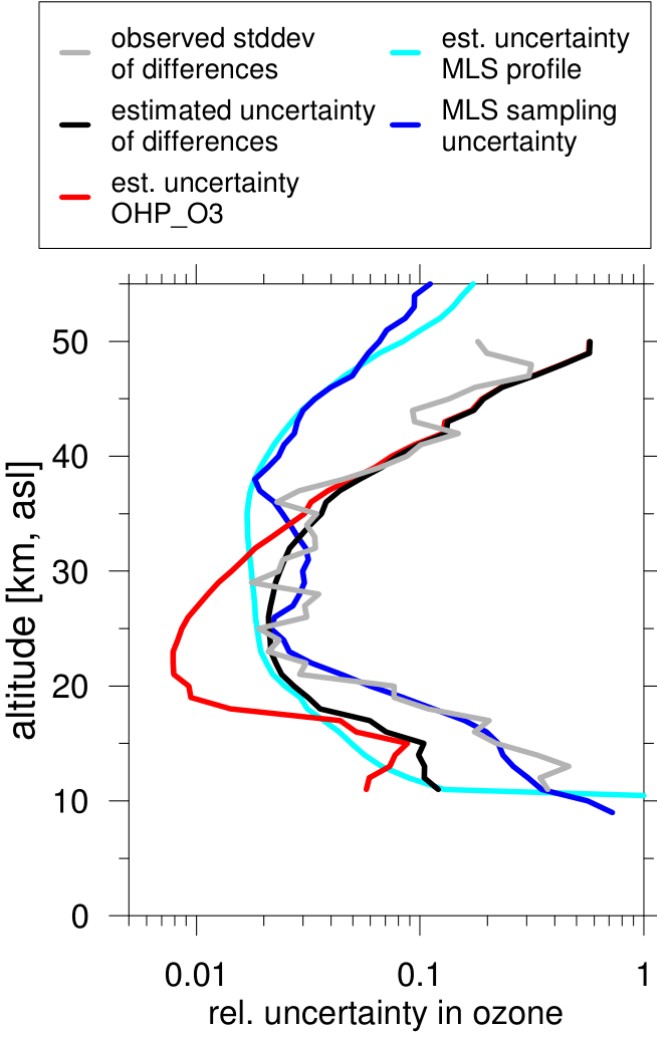

**Figure 12.** Vertical profiles of estimated relative ozone uncertainties for individual MLS profiles (Froidevaux et al., 2008) from the MLS data files (cyan line), and MLS spatial variation / sampling uncertainty estimated from all profiles in the colocation box (blue line). LiO$_3$S ozone uncertainty (red line). Estimated uncertainty for the relative ozone difference (MLS minus LiO$_3$S) based on MLS individual profile uncertainty is given by the black line. The grey line gives the observed standard deviation of the relative ozone differences between MLS and LiO$_3$S during LAVANDE.

Comparing the grey and black lines in Fig. 12, it is obvious that MLS sampling uncertainty (blue line) plays a major role in this intercomparison. From 10 to 30 km it is the dominant source of uncertainty and the major contributor to the observed standard deviation (grey line). Above 35 km, the estimated uncertainty of the LiO$_3$S measurements (red line) is the dominant source of uncertainty - fully consistent with the observed standard deviation (grey line). From Fig. 12 it becomes clear that throughout most of the lower stratosphere, below 25 km, sampling uncertainty (spatial and temporal mis-matches) is a major





limitation for intercomparisons like LAVANDE. To narrow down uncertainties, closer matches and / or a much larger number of coincident events are needed.

Similar results can be seen for SABER ozone profiles in Fig. 13. Again, SABER sampling uncertainty (purple line) domi-nates the uncertainty budget in the relative ozone differences when compared to the LiO$_3$S between 20 and 35 km and needs

5  to be considered to explain the observed standard deviation of the relative ozone differences (grey line). Above 35 km, the uncertainty in the ozone differences is again dominated by the uncertainty of the LiO$_3$S ozone profiles (red line). Also above 35 km, estimated SABER ozone profile uncertainty (pink line) is much smaller than the observed SABER sampling uncertainty (purple line). With the limited number of coincident measurements available during LAVANDE it was, however, not possible to check if this small SABER uncertainty estimate (pink line) is realistic, or too optimistic.



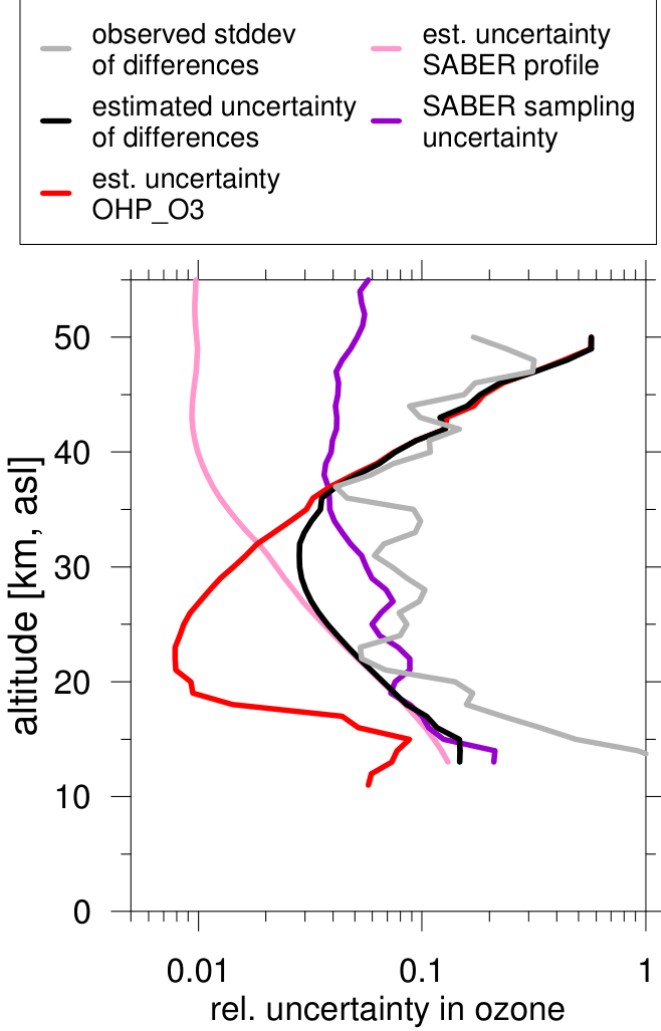

**Figure 13.** Vertical profiles of estimated relative ozone uncertainties for individual SABER profiles (pink line, from Rong et al., 2009), SABER spatial variation / sampling uncertainty over the colocation box (purple line), and LiO$_3$S ozone uncertainty (red line). Estimated uncertainty for the relative ozone difference SABER minus LiO$_3$S based on SABER individual profile uncertainty is given by the black line. The grey line gives observed standard deviation of the relative ozone differences between SABER and LiO$_3$S.

## 5 Intercomparison Results for Temperature

Similar to the analysis done in Fig. 6 for ozone, Fig. 14 shows examples for the temperature time series recorded by the different systems during LAVANDE. As was the case for ozone, a seasonal variation is apparent in the temperature profiles between the two different periods of July 2017 and March 2018. In the upper stratosphere, above 30 km, temperatures were colder in March 2018 than in July 2017, whereas, in the mesosphere, above 70 km, temperatures were colder in July 2017.





All the LAVANDE instruments track these expected seasonal variations. Shorter term variations, such as the slight temperature oscillation appearing near 70 km during July 2017 are also tracked by all the instruments. Near 30 km, some of the NASA-STROZ data points after July $22^{nd}$ seem to lie outside of the usual range, but the temperatures at higher altitudes are consistent with data points from the LTA. This indicates that NASA-STROZ generally provides correct correct temperature profiles, but may have experienced a slight misalignment in a couple of nights.

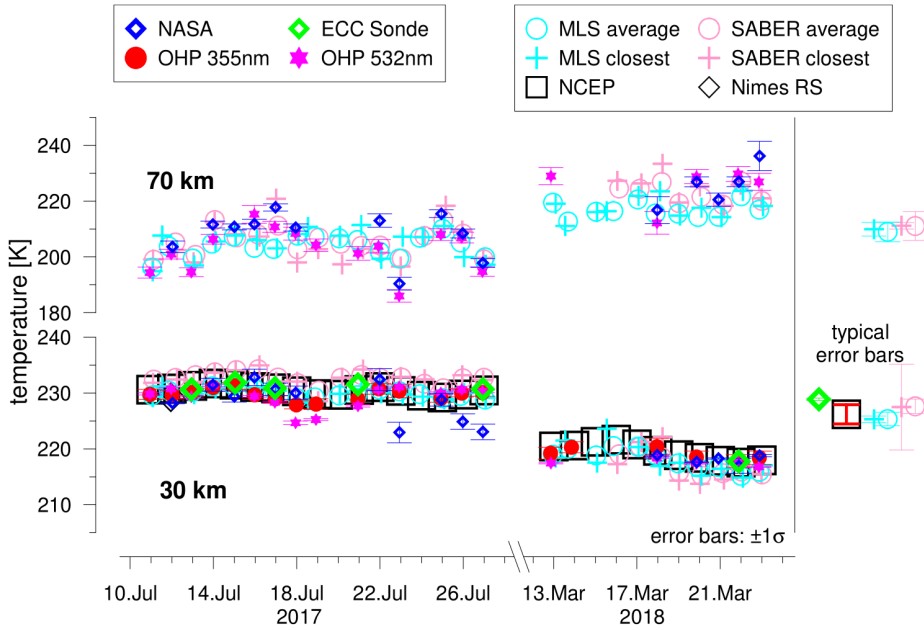

**Figure 14.** Time series of the temperatures measured by the different systems for selected altitude levels during LAVANDE.

The average temperature difference between the various systems and NASA-STROZ is presented in Fig. 15. Unlike the ozone analysis, where the LiO$_3$S was chosen as the reference, NASA-STROZ was chosen here as the reference for temperature, because it had measurements in nearly all nights, and covered a wider altitude range for temperature than either the LiO$_3$S or the LTA. For most altitudes between 25 km and ≈70 km, the agreement between the temperatures from the different LAVANDE systems and temperature from NASA-STROZ is better than ±2 K. Below about 35 km, temperatures from the LiO$_3$S (red), Nîmes radiosondes (yellow), the radiosondes coupled to the OHP ECC sondes (black), and NCEP analyses (cyan) are very similar, indicating that temperatures from NASA-STROZ might be too low by 1 to 4 K in this altitude range. The pronounced increasing cold bias of the LTA data below 30 km arises from signal contamination by aerosols in the lower stratosphere. This bias is less evident in NASA-STROZ and LiO$_3$S as these two lidars operate in the UV at 355 nm as opposed to LTA which operates in the visible at 532 nm and is more susceptible to contamination by aerosol scattering. Above 60 km, LTA (green), SABER (blue), and MLS (magenta) report lower temperatures than those provided by NASA-STROZ. It appears that NASA-STROZ might have a slight warm bias in the upper stratosphere and lower mesosphere, with respect to LTA, which gradually reaches 5 K near 80 km. Warm biases at the top of the lidar temperature profile are commonly associated with errors induced





by the a priori used to initialize the lidar temperature calculation at the topmost levels, or by underestimation of the background (Wing et al., 2018; Sica and Haefele, 2015).

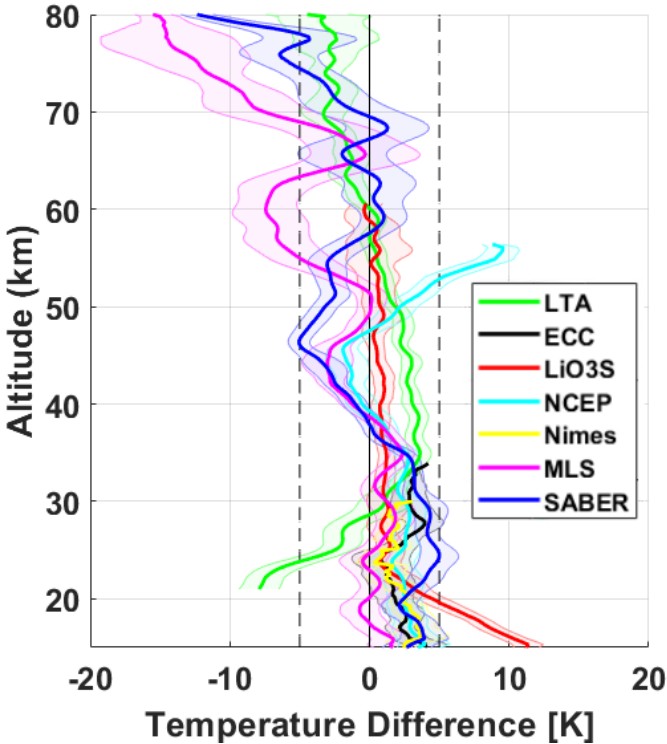

**Figure 15.** Average absolute difference profile between the temperature measured by the various LAVANDE instruments and temperature measured by NASA-STROZ. The shaded range gives ±2 standard deviations of the mean, and indicates statistical uncertainty at the 95% uncertainty level. Results for MLS and SABER are for the weighted average profiles, but very similar results are obtained using the closest match SABER or MLS profiles.

Several other interesting features appear in the temperatures difference profiles at mid-altitudes:

– the higher temperatures reported by the LiO$_3$S below 22 km with respect to the other measurements;

5      – the higher temperatures between 30 and 55 km reported by the LTA. Compared to NASA-STROZ, the LTA reports about 2 K higher temperatures near 40 km and 2 K lower temperatures near 70 km in Fig. 15. Interestingly, this is almost the exact opposite of the difference found between the same two systems in the July 1997 OTOIC intercomparison (Braathen et al., 2004). In OTOIC, NASA-STROZ reported about 2 K higher temperatures than the LTA near 40 km, and 2 K lower temperatures near 70 km. On the other hand, the ≈ 1 K higher temperatures between 35 and 50 km from the

10      LTA compared to the LiO$_3$S during LAVANDE in Fig. 15 are generally consistent with the similar, but slightly smaller, difference found between the same two systems over the 20 year period from 1993 to 2013 by Wing et al. (2018).





- the already mentioned lower temperatures reported by the LTA below 30 km. These are attributed to the much more significant contamination by aerosol scattering at the 532 nm wavelength used by this lidar (compared to 355 nm, used by the other lidars);

- the lower temperatures near 43 km and higher temperatures above 50 km provided by the NCEP analyses;

MLS and SABER temperatures stand out from the ground-based temperature observations as the temperatures exhibit oscillating biases between 35 and 80 km that can reach up to -5 K. A similar oscillating bias for MLS temperatures compared to the OHP lidars (-4 to -6 K near 42 km and near 60 km, no bias near 50 km) was also seen in the 2004 to 2018 long-term intercomparison by Wing et al. (2018b). The same study also found an 'S-shaped' bias for SABER temperatures which also appears in Fig. 15. There SABER temperatures have a warm bias compared to the three temperature lidars below 30 km, and a
cold bias between 40 and 50 km. Wing et al. (2018b) attributed a substantial part of these satellite temperature biases to altitude shifts introduced by the satellite retrieval algorithms.

Examining the scatter of the LAVANDE instrument temperatures in three different altitude regimes yields more detail about the relative biases of each instrument. The left panel of Fig. 16 compares the LAVANDE temperatures from 12 to 35 km to NASA-STROZ. We can see that LTA (green) has a clear aerosol-induced cold bias in the lower half of the panel as it is
systematically colder than every other measurement. We can also see that most data points for the other instruments are below the black reference line indicating that in this altitude range NASA-STROZ reported reliably colder temperatures. The central panel of Fig. 16 represents measurements from 35 to 60 km and exhibits tight correlation between all measurements except MLS. As was noted in Fig. 15, MLS (magenta) has an oscillation in the sign of the temperature bias with respect to the other measurements which is seen here as increased scatter. We can also see the cold bias of NCEP (cyan) in the upper stratosphere.
The right panel of Fig. 16 represents measurements from 60 to 80 km and includes only NASA-STROZ (refernce), LTA (green), MLS (magenta), and SABER (blue). There is generally good tacking between the two lidars with larger scatter for MLS and SABER. We can see some evidence that NASA-STROZ is warmer than the other measurements but not on all nights.





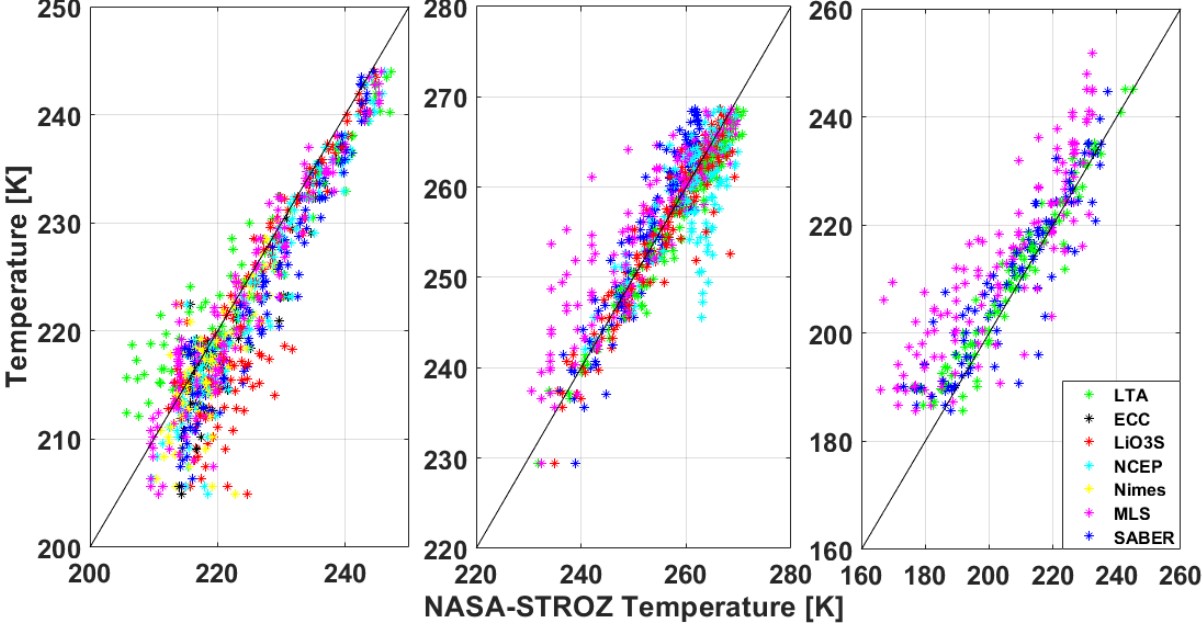

**Figure 16.** Scatter plots of temperature as measured by the various LAVANDE instruments (along the vertical axis) and temperature measured by NASA-STROZ (along the horizontal axis). Left: Temperature from 12 to 35 km altitude. Centre: Temperature from 35 to 60 km altitude. Right: Temperature from 60 to 80 km altitude.

The temperature correlation plot in Fig. 17 shows to what extent temperatures reported by the various systems track the temperature variation measured by NASA-STROZ. The highest correlations, $\geq 0.8$, are seen below 35 km and above 55 km for LTA. Correlations drop significantly near 25 km and again around 50 km which corresponds to regions just above the tropopause and around the stratopause. Similar to the case for ozone in Fig. 9, these drops are associated with small temperature variance at these altitudes, where temperature changes little with altitude, and night to night temperature variations are also small. Measurement noise / uncertainty then becomes prominent and decreases correlations.

Other points to note include: 1) the correlation between the NASA-STROZ and OHP temperature profiles increases again above 50 km and exceeds 0.9 between 60 and 80 km. 2) lower correlation is seen for temperature from the $LiO_3S$ above 50 km. This is likely caused by increasing measurement uncertainty for temperature from the $LiO_3S$ above 55 km which is associated with the lower laser output at 355 nm in this system. The 355 nm Nd:Yag energy output in $LiO_3S$ is intentionally reduced by manually introducing delay in the laser oscillator. This is done to optimise the system for comparison with the 308 nm laser signal. 3) MLS and SABER temperatures show lower correlation with respect to all other instruments. 5) Excluding the region associated with the tropospheric temperature minimum, the correlation between NASA-STROZ temperatures and the onsite ECC sondes, Nîmes radiosondes (up to 30 km), and NCEP analyses (up to about 40 km) is also good. Above 40 km, in the topmost NCEP analysis pressure levels at 1 and 0.4 hPa ($\approx$48 and 54 km), correlation drops rapidly for the NCEP analyses.





This has also been seen in previous intercomparisons (e.g., Steinbrecht et al., 2009b). At these top-levels, the NCEP analyses are relaxed substantially towards a climatological state, and are much less responsive to actual temperature variations.

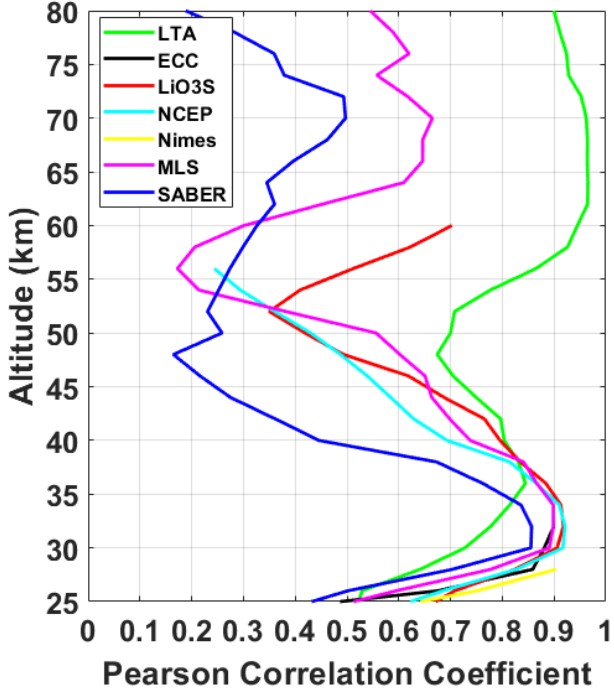

**Figure 17.** Vertical profiles of the correlation between temperatures reported by the various LAVANDE systems and temperature measured by NASA-STROZ. Outliers were excluded. Correlation is taken over the 28 nights of LAVANDE, and over 2 kilometres in altitude. For MLS and SABER correlations are given for the weighted average profiles.

### 5.1 Lidar Temperature Uncertainty Analysis

A closer look at temperature measurement uncertainties is taken in Figs. 18 to 21. The approach in this section is the same as for

5   ozone in the previous section. Fig. 18 shows the estimated temperature uncertainty for NASA-STROZ (blue) and LiO$_3$S (red). The largest term contributing to the total uncertainty for lidar temperatures below 80 km comes from the Poisson statistics of the limited number of photons scattered back from high altitudes (see e.g., Leblanc et al., 2016c; Sica and Haefele, 2015). The temperature uncertainty for NASA-STROZ is estimated to be less than 1 K between 15 and 50 km, increasing to 4 K near 80 km, very similar to the comprehensive uncertainty given for a typical stratospheric lidar in Fig. 10 of Leblanc et al.

10   (2016c). For the LiO$_3$S, temperature uncertainty is also estimated to be less than 1 K below 30 km, but increases to 10 K near 60 km. From these two uncertainties, the uncertainty of the difference between coincident temperature profiles from LiO$_3$S and NASA-STROZ can be estimated (similar to what was discussed in the previous section for ozone). This estimated uncertainty of the temperature difference is shown by the black line in Fig. 18.



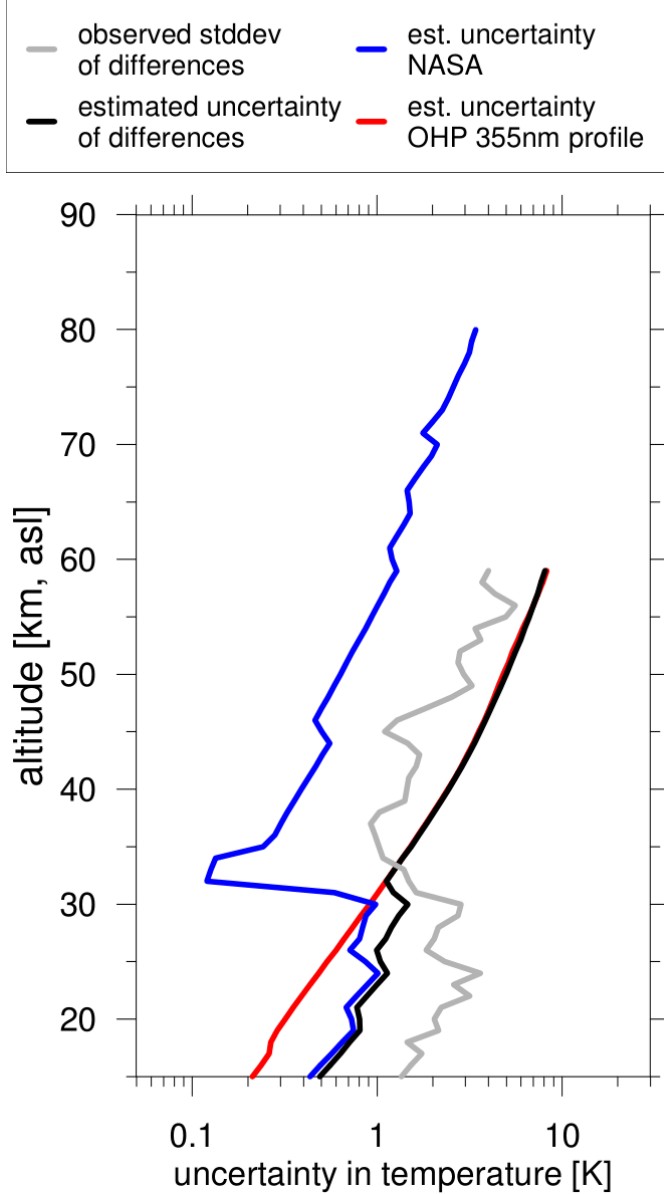

**Figure 18.** Similar to Figs. 10 to 13, but for temperature. Plotted are: estimated uncertainty for temperature measured by LiO₃S (red curve) and NASA-STROZ (blue curve), estimated uncertainty for temperature differences between the two systems (black curve), and observed standard deviation of temperature differences between the two systems (grey curve) during LAVANDE.

If the estimated uncertainties for the two lidars are correct, the black line in Fig. 18 should be very similar to the grey line, which shows the observed standard deviation of the temperature difference between LiO₃S and NASA-STROZ over all the





(nearly coincident) measurements during LAVANDE. Unfortunately, the agreement between the black and grey curves is not so good in Fig. 18. Above 30 km, the observed standard deviation is actually smaller than the estimated uncertainty, by a factor of about 2. This indicates that the estimated temperature uncertainty for the $LiO_3S$ is too large above 30 km, by a factor of about 2. This may arise from incorrect accounting for the vertical integration and filtering of the temperature profile in the uncertainty

estimate for the $LiO_3S$. On the other hand, below 30 km, the observed standard deviation (grey line) is larger than the estimated uncertainty (black line), again by a factor of about 2. This would indicate that the estimated temperature uncertainty for $LiO_3S$ and/or NASA-STROZ is too small, by a factor of 2 or more. It could mean that other sources of uncertainty, beyond statistical uncertainty, are important. Future work will be conducted using the results of this intercomparison campaign to refine the $LiO_3S$ error budget for temperature.

The corresponding comparison of uncertainties for LTA and NASA-STROZ are given in Fig. 19. Both systems have very similar estimates of temperature uncertainty, which are also consistent with the recommendations of Leblanc et al. (2016c). Above 60 km, the estimated uncertainty of the temperature difference (black curve) is similar with the observed standard deviation during LAVANDE (grey curve), confirming the uncertainty estimates for the two lidars above 60 km. However, at most altitudes below 60 km, the observed standard deviation (grey curve) remains at 2 to 3 K. This is substantially larger, by

up to a factor of 10, than the estimate (black curve).

This indicates that the uncertainty estimates for LTA and NASA-STROZ (and also the temperature estimate in Fig. 10 of Leblanc et al., 2016c) are too optimistic during LAVANDE. Detector misalignment in one or both lidars is likely the main cause of the reported disagreement. At OHP the alignment is made manually each night by operators and a slight misalignment may induce a detectable temperature bias. Given that even a small 1% error in the slope of the density profile can induce a 2

to 2.5 K bias in the resulting temperature profile, the possibility of human errors exists. A key conclusion from this study is that automatic alignment systems for NDACC lidars are essential for measurement accuracy and long-term stability. Another source of error may come from the linearisation correction of the photon counting at high counting rate.



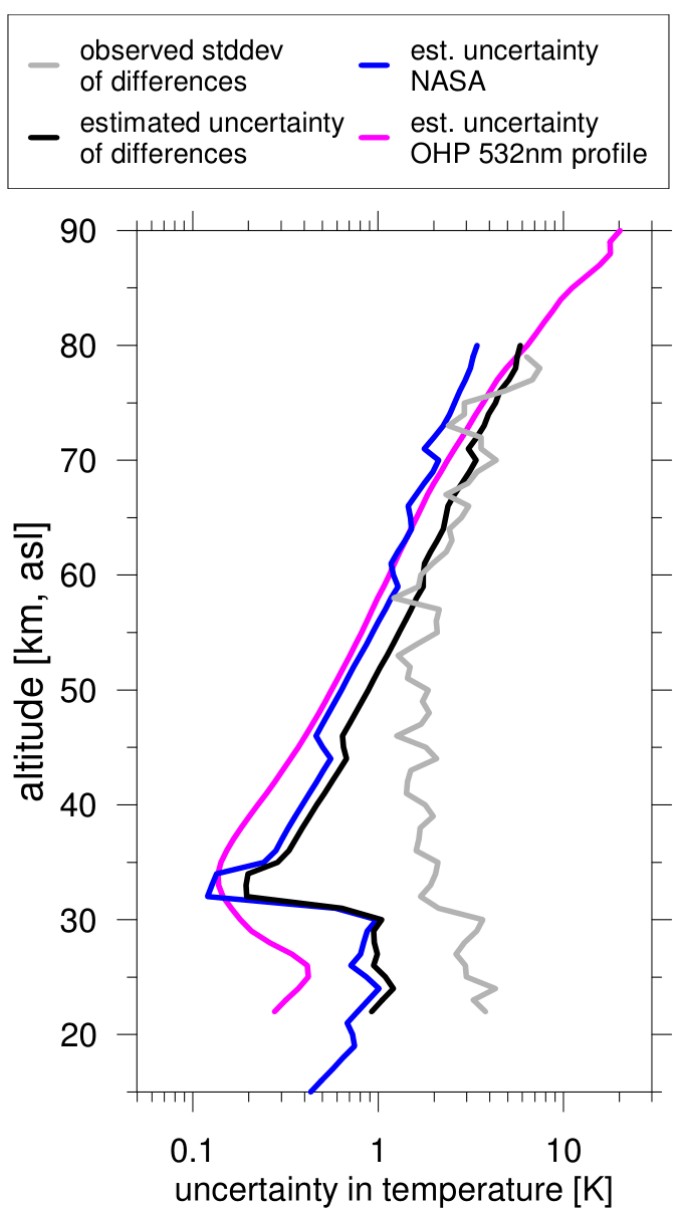

**Figure 19.** Same as Fig. 18, but for LTA (magenta curve) and NASA-STROZ.

## 5.2 Satellite Temperature Uncertainty Analysis

In the next section we extend the comparison of uncertainty estimates and observed difference standard deviation to temperature profiles from the MLS and SABER satellite instruments. As with ozone, temporal and spatial mismatch between the lidar





measurement at OHP and the number of satellite measurements within the chosen coincidence box (see Fig. 1) plays an important role. Fig. 20 allows comparison of the single profile uncertainty given for the MLS data (cyan line) with the estimated sampling uncertainty for the weighted mean MLS profile (light blue curve). Sampling uncertainty is estimated by the weighted standard deviation of all MLS profiles in the coincidence box (which implicitly includes single profile uncertainty). Clearly,

5  for MLS sampling uncertainty is larger than single profile uncertainty (e.g. Schwartz et al., 2008), by a factor of about 2. Sampling uncertainty is also the dominating uncertainty for the MLS minus NASA-STROZ temperature differences observed in LAVANDE (grey curve). When sampling uncertainty is included in the estimate for total temperature difference uncertainty (black curve in Fig. 20), good agreement is obtained with the observed standard deviation (grey curve). This good agreement would not be achieved, if only the MLS single profile uncertainty would be considered (cyan line). Then the corresponding

10  estimated temperature difference uncertainty would be too small. Overall, Fig. 20 confirms that 1) MLS single profile temperature uncertainty is of the order of 1 to 3 K; 2) NASA-STROZ provides comparable single profile uncertainty (1 to 3 K); and 3) sampling uncertainty plays an important role in the total uncertainty budget for the satellite vs. ground-based intercomparison, contributing an uncertainty of 2 to 5 K during LAVANDE.



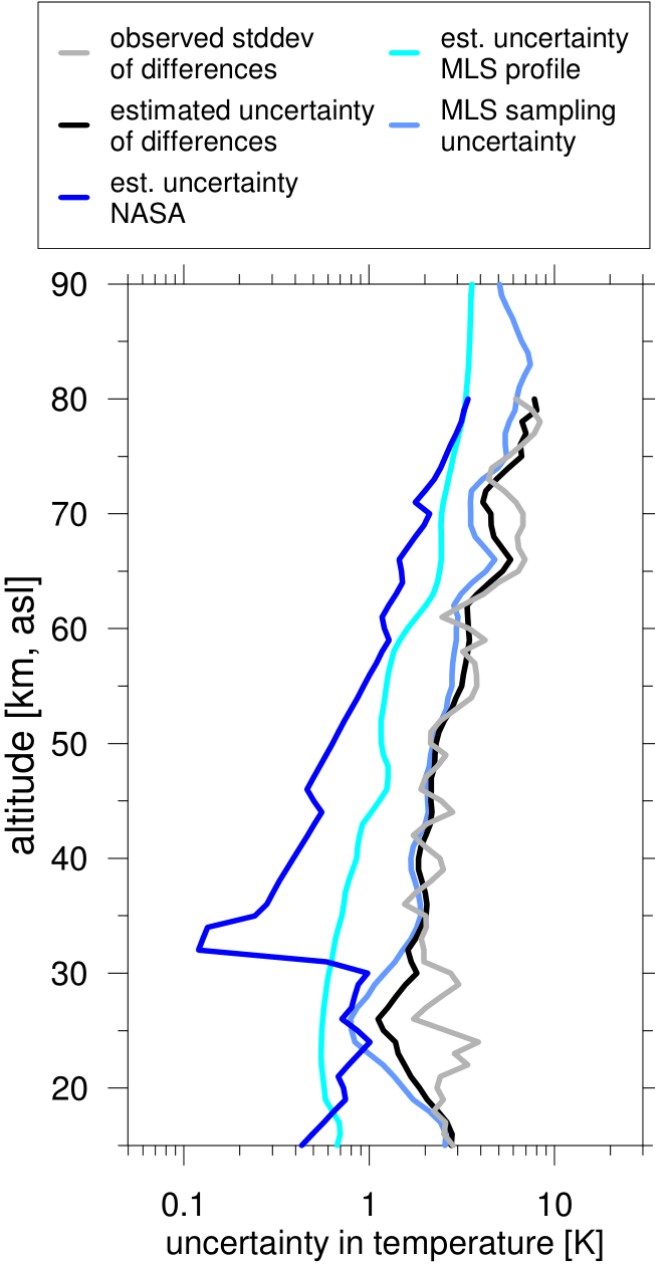

**Figure 20.** Same as Figs. 18 and 19, but comparing single profile and sampling uncertainties of MLS satellite temperature profiles (cyan and light blue curves) and NASA-STROZ ground-based profiles. Results are for the MLS weighted average profiles, but very similar results are obtained for closest match MLS profiles. MLS single profile uncertainty is included in the data distribution and is described in Schwartz et al. (2008)



Similar results are obtained in Fig. 21 for SABER temperature profiles. Also for SABER, sampling uncertainty (purple curve) is larger than single profile uncertainty (pink curve, estimated following Rezac et al., 2015a, b). Sampling uncertainty must, again, be considered to explain the observed standard deviation of SABER - NASA-STROZ temperature differences (black curve matching grey curve). Below 40 km, however the observed standard deviation (grey) is about 2 K larger than estimated from SABER sampling uncertainty and NASA-STROZ temperature uncertainty. Similar disagreement below $\approx$40 km was already mentioned for the $LiO_3S$ vs. NASA-STROZ comparison in Fig. 10, and for the LTA vs. NASA-STROZ comparison below 50 km in Fig. 19. Similar underestimation occurs when the $LiO_3S$ or LTA are used as reference instead of the temperature from NASA-STROZ. So this disagreement is fairly consistent across all the lidar instruments. All this indicates again that the very small temperature uncertainty estimates of less than 1 K below 50 km for NASA-STROZ and other lidars are too optimistic. Additional uncertainty sources not considered in Leblanc et al. (2016c) may play a role (e.g. temporal changes in alignment, defocusing, multiple scattering etc.). From the LAVANDE results shown in Figs. 10 to 21 is seems that a temperature uncertainty of 1 to 3 K also below 50 km is not unrealistic for the participating lidar systems.





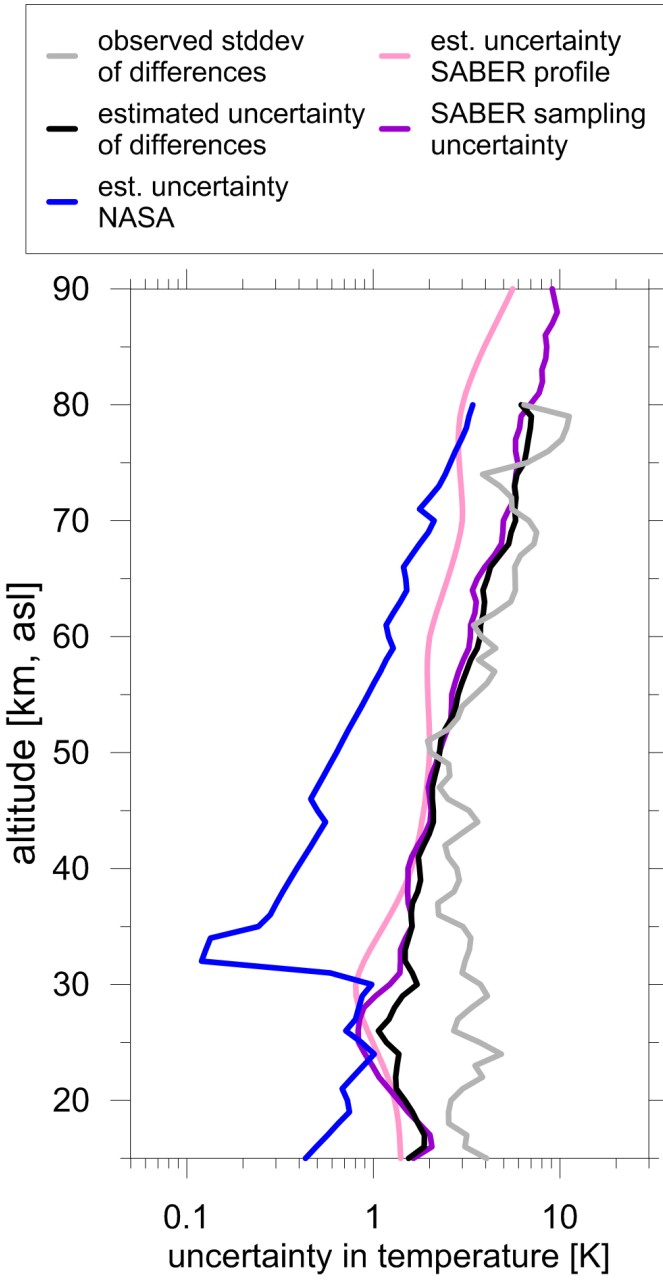

**Figure 21.** Same as Fig. 20, but for SABER satellite temperature profiles (pink and purple curves) and NASA-STROZ. Results for SABER are shown for the weighted average profiles, but very similar results are obtained for closest match SABER profiles. SABER single profile temperature uncertainty was estimated following Rezac et al. (2015a, b).





## 6 Conclusions

The LAVANDE intercomparison of the OHP lidars (tropospheric DIAL, stratospheric DIAL, and Rayleigh temperature), local radiosondes and ECCs, satellite instruments MLS and SABER, and the mobile NDACC reference lidar NASA-STROZ has shown overall good tracking of both vertical profiles of temperature and ozone for all participating instruments. LAVANDE

was a "blind" intercomparison, i.e. all ground-based measurements presented here were submitted "blind". There was no possibility to see results from the other instruments before submitting each groups data.

Agreement for ozone was within ± 10% for all instruments between approximately 15 and 40 km. Agreement was closer, better than ± 5%, between 18 and 38 km for the two stratospheric DIAL systems. Some statistically significant differences are present in the two stratospheric systems when measuring low ozone densities below 14 km and above 40 km. The tropospheric

DIAL, $LiO_3T$ also reported lower ozone concentrations than the local ECC and than $LiO_3S$ above 10 km (bias > 10%). Although this may improve with further corrections of the ECC in the stratosphere, it is related to the increasing uncertainty of the $LiO_3T$ near its upper measurement range. Improvement of the lidar data processing and removal of this potential bias will be investigated in future work involving Optimal Estimation Techniques (Farhani et al., 2019). Future tropospheric ozone lidar campaigns for NDACC lidars would be required to assess the new technique and fully characterise any residual biases. MLS

and SABER ozone profiles agree with the profiles produced by lidars and ECCs from about 20 km to above 40 km. Below 20 km, both sets of satellite profiles deviate significantly from the lidars and the ECCs. Above 40 km, ozone measurement uncertainties become large for the lidars, and differences increase while their significance goes down.

The assessment of the uncertainty budget for ozone concentration profiles for each instrument showed that the reported uncertainties for both $LiO_3S$ and NASA-STROZ are well characterised and realistic. The reported uncertainty estimates for

ECCs from Tarasick et al. (2016) appear too optimistic for the sondes launched during LAVANDE. They seem to underestimate the total uncertainty for the LAVANDE ECC sondes by a factor of 2. When comparing the ground-based profiles to the satellite measurements it is necessary to account for sampling uncertainty, i.e. real ozone differences between the ground-based profile and the satellite profiles measured a couple of hundred kilometers and a few hours away. This sampling uncertainty for MLS was greater than the reported single profile uncertainty below 30 km and dominates the error budget in this region. For SABER,

sampling uncertainty is substantially larger than single ozone profile uncertainty at all altitudes above 30 km. Above 35 km, MLS and SABER sampling uncertainty was less relevant, because lidar ozone uncertainties become larger.

Agreement for temperature was within ±5 K for all instruments between approximately 25 and 80 km. Below 30 km, the LTA operating at 532 nm has a well-known aerosol-induced cold bias relative to the other instruments. This bias will be corrected in the future with the installation of a rotational Raman channel for lower atmospheric temperatures. The $LiO_3S$

reports significantly higher temperatures below 23 km, which will be corrected in future data releases. NASA-STROZ has an apparent warm bias above 70 km, likely due to a priori assumptions or background estimations made in the profile retrieval. Radiosondes and ECCs are in good agreement with the lidar profiles. MLS has a pronounced oscillating temperature bias throughout the middle atmosphere. SABER has a slight cold bias near the stratopause (45 km). Both of these biases are consistent with altitude distortions in the satellite retrieved altitude grid (see also Wing et al., 2018b).





The assessment of the uncertainty budget for temperature profiles showed that the reported uncertainties for the LiO$_3$S may be underestimated below 30 km and overestimated at higher altitudes. Both the LTA and NASA-STROZ appear to underestimate the total uncertainty in the temperature profiles below 55 km. This may indicate that other sources of uncertainty, beyond those in Leblanc et al. (2016c), may need to be considered. When comparing ground-based the temperature profiles with satel-

5    lite measured profiles from MLS and SABER it is necessary to include sampling uncertainty. For MLS, sampling uncertainty during LAVANDE was between 2 and 8 K, about 2 times larger than single profile uncertainty at most altitudes from 20 to 80 km. Similar sampling uncertainty was found for temperature profiles measured by SABER during LAVANDE.

Overall, the LAVANDE campaign has successfully validated the NDACC lidar profiles for both temperature and ozone over a large vertical extent. We have identified a few minor biases existing at both the low and high limits of our profiles, which

10    we shall address going forward. Additionally, we have shown that sampling uncertainty can dominate reported uncertainty in lidar-satellite comparisons and that NDACC temperature lidars have a larger variance below 50 km than can be explained solely by statistical uncertainties.



**Table 1.** Instruments compared during the LAVANDE campaign in July 2017 and March 2018.

| Instrument | Measurement of ozone | Altitude range | Measurement of temperature | Altitude range | Data source |
|---|---|---|---|---|---|
| NASA-STROZ | DIAL (308 and 355 nm) | 10 to 50 km | Rayleigh and Raman lidar (355 nm) | 10 to 70 km | [1] |
| LiO$_3$S | DIAL (308 and 355 nm) | 10 to 50 km | Rayleigh lidar (355 nm) | 25 to 60 km | [1] |
| LiO$_3$T | DIAL (289 and 316 nm) | 2.5 to 13 km | | | [1] |
| LTA | | | Rayleigh lidar (532 nm) | 30 to 80 km | [1] |
| OHP ECC sondes (ENSCI-Z) | KI electro chemical cell | 0 to 35 km | Thermistor (Modem M10) | 0 to 35 km | [1] |
| Nîmes radiosondes | | | Thermistor (Modem M10) | 0 to 35 km | [2] |
| NCEP analyses | | | Meteorological data assimilation | 0 to 50 km | [3] |
| MLS satellite, Version 4.23 | $\mu$wave limb sounding (240 GHz) | 10 to 80 km | $\mu$wave limb sounding (118 GHz) | 15 to 90 km | [4] |
| SABER satellite, Version 2.0 | IR limb sounding (9.6, 1.27 $\mu$m) | 15 to 90 km | IR limb sounding (4.3, 15 $\mu$m) | 10 to 100 km | [5] |





**Table 2.** Measurement dates for the ground-based instruments during the LAVANDE campaign in July 2017 and March 2018. The lidar measurements require night-time conditions and averaging over several hours. The dates give the beginning of these nights. X denotes a valid measurement for the given night. (x) denotes a measurement that appeared faulty and was not used in the later statistical analysis. Satellite profiles of ozone and temperature are available for all nights.

| start of night | ozone | | ozone & temp. | | total $O_3$ | temperature only | | | | total temp. |
| | OHP tropo | NASA[a] STROZ | OHP[b] DIAL | ECC[b] sonde | | NASA[a] temp | OHP temp | Nîmes sonde | NCEP[c] anal. | |
|---|---|---|---|---|---|---|---|---|---|---|
| July 10 | X | | X | | 2 | | X | X | X | 4 |
| July 11 | | X | X | X | 3 | X | X | X | X | 6 |
| July 12 | X | (x) | X | X | 3 | (x) | X | X | X | 5 |
| July 13 | | X | X | | 2 | X | X | X | X | 5 |
| July 14 | X | X | X | X | 4 | X | | X | X | 5 |
| July 15 | | X | X | | 2 | X | X | X | X | 5 |
| July 16 | X | X | X | X | 4 | X | X | X | X | 6 |
| July 17 | X | X | X | | 3 | X | X | X | X | 5 |
| July 18 | X | | X | | 2 | | X | | X | 3 |
| July 19 | | | | | 0 | | | X | X | 2 |
| July 20 | X | | X | X | 3 | | X | X | X | 5 |
| July 21 | | | X | | 1 | X | X | X | X | 5 |
| July 22 | X | | X | X | 3 | X | X | X | X | 6 |
| July 23 | | | | | 0 | | | X | X | 2 |
| July 24 | | | X | | 1 | X | X | X | X | 5 |
| July 25 | X | | X | | 2 | X | X | X | X | 5 |
| July 26 | X | | X | X | 3 | (x) | X | X | X | 5 |
| subtotal | 10 | 6 | 15 | 7 | 38 | 10 | 14 | 16 | 17 | 79 |
| March 12 | | | X | | 1 | | X | X | X | 4 |
| March 13 | | | X | X | 2 | | | X | X | 4 |
| March 14 | | | | | 0 | | | X | X | 2 |
| March 15 | | | | | 0 | | | X | X | 2 |
| March 16 | | | | | 0 | | | X | X | 2 |
| March 17 | | X | X | | 2 | X | X | X | X | 5 |
| March 18 | | | | | 0 | | | X | X | 2 |
| March 19 | X | X | X | X | 4 | X | X | X | X | 6 |
| March 20 | X | X | X[d] | (x) | 3 | X | X | X | X | 5 |
| March 21 | X | X | X | X | 4 | X | X | X | X | 6 |
| March 22 | X | X | X | X | 4 | X | X | X | X | 5 |
| subtotal | 4 | 5 | 7 | 4 | 20 | 5 | 6 | 11 | 11 | 43 |
| grand total | 14 | 10 | 22 | 11 | 58 | 15 | 20 | 27 | 28 | 122 |

[a] Due to a laser failure on July 18[th] the NASA-STROZ system was not able to measure ozone profiles for the rest of July 2017. Temperature measurements were still possible and a separate column was included for temperature profiles from the NASA system.

[b] the $LiO_3S$ system and the ECC sondes measure both ozone and temperature profiles.

[c] NCEP analyses usually provide data for 12 UTC. For comparison with the nightly mean lidar profiles (typically around 20:30 UTC) we used the average of the two 12 UTC analyses before and after each night.

[d] the $LiO_3S$ temperature profile was clearly faulty on that night, but the ozone profile appeared to be fine.



*Data availability.* The data that support the findings of this study are openly available in the [1] The data used in this publication were obtained from LATMOS as part of the Network for the Detection of Atmospheric Composition Change (NDACC) and are publicly available ftp://ftp.cpc.ncep.noaa.gov/ndacc/station/ohp/, last access: 17 June 2018 [2] local radiosondings from Nîmes http://weather.uwyo.edu/upperair/sounding.html, last access: 10 June 2018 [3] NCEP model profiles ftp://ftp.cpc.ncep.noaa.gov/ndacc/ncep/temp/, last access: 1 June 2018 [4] MLS temperature and ozone profiles https://disc.gsfc.nasa.gov/datasets?keywords=MLS, last access: 17 June 2018 and [5] SABER temperature and ozone profiles http://saber.gats-inc.com/, last access: 17 June 2018

*Author contributions.* RW, SGB, TJM, JTS, and GS conducted the measurement campaign at OHP. WS conducted the blind comparison of all LAVANDE data. RW and WS drafted the article. AH, SGB, SK, GA, PK, and TJM provided access to the data and instruments. All authors discussed the results and contributed to the final paper.

*Competing interests.* The authors declare that they have no conflict of interest.

*Acknowledgements.* This work is supported by Institut National des Sciences de l'Univers/Centre National de la Recherche Scientifique (INSU/CNRS), Université de Versailles Saint-Quentin-en-Yvelines (UVSQ), Centre National d'Études Spatiales (CNES), the NASA Upper Atmospheric Research Program, and ARISE2. The authors would particularly like to thank the technicians at La Station Géophysique Gérard Mégie at OHP, who are so important for running the long-term measurement program.



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
