# Peer review of "Intercomparison and Evaluation of Ground- and Satellite-Based Stratospheric Ozone and Temperature profiles above Observatoire de Haute Provence during the Lidar Validation NDACC Experiment (LAVANDE)"

_Atmospheric Measurement Techniques, 2020_

## Referee Comment (RC1) · Anonymous Referee #1 · 5 Jun 2020

This manuscript presents the results of a blind inter-comparison campaign that took place at the Observatoire de Haute Provence (OHP), France, a well-known long-term atmospheric composition monitoring station of NDACC. The results cover the ozone and temperature measurements of 3 lidars permanently deployed at OHP (LTA, LiO3S and LiO3T), the mobile lidar STROZ from NASA-GSFC deployed at OHP for the occasion, co-located ECC ozonesondes, nearby radiosondes, and coincident satellite measurements from Aura-MLS and SABER.

[Figure]

These inter-comparison campaigns are essential to characterize the performance of the ground-based instruments, often considered "reference measurements" when validating satellite-borne instruments, and when long-term intercalibration between these satellite measurements is needed. This, together with the intrinsic value of the OHP timeseries themselves, makes the publication of these results in AMT highly relevant.

The take-home messages, as written in the abstract and conclusion, are clear and provide a good basis for reference in the future use of these datasets. Overall, the methods used are appropriate, but in several instances, a lack of clarity or rigor casts some doubts on the validity of some of the results, or more importantly, their interpretation. Two examples are 1) the loose/ambiguous reference to uncertainty and how it is used in the manuscript, and 2) the comparison of lidar and satellite uncertainty estimates and resulting conclusions. For this reason, major revisions are recommended before the manuscript can be considered for AMT publication. My comments and suggestions (major and minor points) are included below.

Major points:

1) Historically, the ozonesondes have typically been considered "independent" measurements. The ozone correction, as described here (i.e., using SAOZ), makes them dependent on the SAOZ measurement as well as the balloon blasting altitude. Recently, there has been a global effort for ozone sounding homogenization worldwide led by the SHADOZ community. Does this effort apply to the French ozonesonde program? Was the French ozonesonde team involved in this effort? Please clarify the role of the ozonesondes: Are they considered reference or just correlative measurements?

2) It is difficult to figure out what the authors refer to as "uncertainty". For example, Page 17, line 9-10, "average measurement noise" is mentioned, and then in the same sentence "standard deviation of the ozone difference". Are the authors referring to the combined uncertainty of the two measurements? If so, please use "combined uncertainty" instead of "standard deviation".

3) Please make the clear distinction between what is random, what is systematic, and how these two types of uncertainty components are treated in the various parts of the manuscript. For example, in Page 20, line 16, it is claimed that the "uncertainty estimates...are too optimistic". Do these estimates account for systematic effects as well (total uncertainty?), or just the random component? If just random, it is not surprising that they do not match the r.m.s. differences, as r.m.s. will also reflect the presence of pseudo-systematic errors (e.g., alignment error for LiO3S, or aerosol interference for LTA).

4) It is not clear what MLS single profile uncertainty is. Please clarify. A single profile uncertainty is used. Aren't several MLS profiles used in the comparisons? Please clarify.

5) The authors' interpretation of Fig 21 is overstated and inconsistent with that of the previous figure. The two sentences starting with "So this disagreement..." on page 34, line 8-9 assume that because disagreement is found for all lidars, then all lidars are "wrong". What if the source of the disagreement originates in SABER's underestimated uncertainty or a systematic error in the SABER profiles? If STROZ uncertainty is underestimated, why don't we see it in the comparison with MLS (Fig 20)?

6) Section 3.1.1: There is no evidence of vertical offset in figures 4 and 5. There is a difference in the shape of the peak between MLS and the others, but this does not seem to be the result of an altitude offset. For example, the MLS and GB ozone profiles are on top of each other at all altitudes above 24 km and all altitudes below 12 km. Also there is no evidence on these figures that SABER ozone/temp and MLS temp are shifted in altitude.

Minor points:

The title should read "Observatoire de Haute Provence Âż

Introduction: the authors should focus less on listing all the past campaigns and more

on explaining the purpose of those campaigns and their outcome (include quantitative results as well as the main take-home messages from these campaigns.

Page 3, Line 3: Add "Aura"

Page 3, Line 29: Replace "off-line" by "non-absorbed"

Page 4, 2.0.2.: The few technical details in this instrument description section do not convey the right message. Please specify that tropospheric DIAL requires more absorbing wavelengths (stronger UV) to measure ozone at ppb levels rather than ppm levels, which is why the wavelengths are different from stratospheric DIAL. Also, specify that the initial 266 nm beam is spectrally shifted by the Raman cell to produce 289 and 316 nm.

Page 4, line 13: Remove "absolute"

Page 4, 2.0.3: As for paragraph 2.0.2., the choice of information included in this paragraph is somewhat arbitrary. There are other corrections applied to the signals to obtain the temperature profile (background noise, dead-time, molecular and particulate extinction). There is also a temperature initialization procedure at the top of the profile. I do not think the range-square correction should be mentioned without mentioning the other effects. I would recommend to add more details, or simply to mention that this is the backscatter temperature lidar technique, obtained by downward integration of atmospheric density (cf. Hauchecorne and Chanin).

Page 5, line 19: Is GPH converted to geometric altitude before it is used for comparison with lidar? Please specify.

Page 8, line 4: The impact of effective vertical resolution mentioned line 13 should be mentioned here

Page 8, sentence starting with "The increased spring time variance": What is the purpose of this sentence? Is it supposed to introduce work further down in this manuscript? Please clarify.

[Figure]

Page 9, line 22: The large percent differences between MLS and the other instruments is unlikely to be associated with MLS vertical resolution. It is mainly because the ozone peak and ozone minimum are registered at different altitudes. MLS is capable of identifying these sharp transitions. The main reason for the observed difference is most likely the spatiotemporal coincidence and atmospheric variability

Page 10, line 3-4: This is inconsistent with the figure. In fact, the best agreement is below 25 km

Page 10, line 8: I do not see any disagreement in the altitude. The peak is just smoother, and SABER actually reproduces well the ozone minimum right below the peak

Page 11, line3: I do not think the addition of a new Raman channel will reduce the warm bias. Please rephrase

Page 13, line 5: Typo

Page 17, line 3: Please clarify. Does "uncertainty estimated by the retrievals" include only random components (photon noise), or is it the total uncertainty? How is this average computed?

Page 17, Equation 1: Define L and N

Page 20, line 8-12: Please define "MLS individual profile uncertainty"? Is that precision (random) or total uncertainty? Shouldn't a "campaign mean" of the individual uncertainties be considered instead of a single profile uncertainty? (just like it was done for GB instruments)

Correlation diagnostics (section 4, page 15 and section 5): What is the purpose of the correlation diagnostic? This diagnostic seems to introduce more confusion than clarification on the origins of the differences between the instruments. For example the authors state that the method is sensitive to the size of the averaging window, "drastically increasing or decreasing the amplitude of this peak". Please clarify or remove

this part to keep the discussions of Figures 7 and 15.

Page 25, line 1: What is the a priori source? What is the altitude of initialization? Is it the same altitude for LTA and STROZ? LiO3S initialization is much lower. What a priori do they use? In order to investigate the STROZ warm bias at the top, could an alternate data processing be done using the same a priori at the same altitude? This would remove any bias associated with the tie-on procedure.

Page 27, line 8, "between 60 and 80 km": This sentence is misleading. If STROZ and LTA use the same a priori source (MSIS?), it is not surprising that the correlation increases as the profiles approach their tie-on altitude. This high correlation does not demonstrate instrument performance.

Page 30, line 7: "beyond statistical uncertainty": Systematic uncertainty components must be included in Figure 18, especially if they are not negligible, for example, uncertainty associated with temperature initialization and background noise correction in the mesosphere, and possibly dead-time correction uncertainty at the bottom of the profiles

Page 30, line 16, "...and also the temperature estimate of fig 10 of Leblanc et al.) are too optimistic..": I do not understand this sentence. Leblanc's figure 10 shows an example of uncertainty budget for a different lidar system (unrelated to LAVANDE), not including the impact of aerosols or misalignment. It is not surprising to find different results here, especially if the LTA profiles are impacted by aerosol and/or misalignment.

Section 5: There is no attempt to explain the NCEP differences (SSU?, AMSU?) Any published reference? Did the authors consider using MERRA-2?

Page 37, line 3, "Other sources of uncertainty": The authors are correct that other sources of uncertainty must probably be accounted for. But they should also discuss the possibility of optimizing the instrument set up (in this case alignment) so that errors are minimized and the introduction of additional uncertainty sources is less relevant.

Figures:

Figures 2-3: are too small.

Figure 4: On the right panel, the differences between the instruments are not shown below 8-10 km. Please plot the differences It would be good to uniformize the instrument short names throughout all figures and text. For example, sometimes, we see "OHP 532 nm", sometimes "LTA".

[Figure]

---

## Referee Comment (RC2) · Anonymous Referee #2 · 19 Jun 2020

Review of "Intercomparison and Evaluation of Ground- and Satellite-Based Stratospheric Ozone and Temperature profiles above Observatoire Haute Provence during the Lidar Validation NDACC Experiment (LAVANDE)" by Wing et al.

The study presents a detailed comparison of the stationary lidars at OHP with a series of other data sets, in particular a mobile lidar system from NASA. Also ECC, MLS, SABER ozone and temperature profiles are considered. The comparison has been

performed "blind" by an impartial expert, which is a very interesting approach. For the comparison, a wide range of visualizations is presented reaching from mean profiles, over time series and scatter plots to correlation profiles. This gives a detailed insight in the behavior of each data set but makes the paper a bit lengthy. The study also evaluates in detail the uncertainties of each data set, which is a very interesting aspect and should be done more in validation studies.

Validations studies such as the one presented here are important contributions to understand instrumental differences and to obtain consistent long term data sets. Methods and results are well explained and the paper follows a logic structure.

I recommend the paper for publication in AMT and provide below minor comments for the authors to consider.

Minor comments

In the effort to identify co-located profiles between lidar and satellite the authors allow a time difference of up to 12 hours. Given the diurnal cycles in temperature and ozone in the stratosphere and mesosphere, this seems too tolerant. Have any effects related to tides and diurnal cycles been corrected? An analysis of the distribution of the time differences would be helpful to convince the reader that systematic biases are not a consequences of diurnal cycles.

The weighted mean on p6, l16 takes into account the typical wind speed in the stratosphere. What is the justification for this? Diurnal cycles in temperature and ozone are not driven by advection but by photochemistry and tidal waves. Please comment.

P6, l11: do you mean 10 to 20 matching profiles per night? With the chosen wording this is not absolutely clear.

---

## Author Comment (AC1) · 8 Jul 2020

**Response R1: "Intercomparison and Evaluation of Ground- and Satellite-Based Stratospheric Ozone and Temperature profiles above Observatoire Haute Provence during the Lidar Validation NDACC Experiment (LAVANDE)"**

This manuscript presents the results of a blind inter-comparison campaign that took place at the Observatoire de Haute Provence (OHP), France, a well-known long-term atmospheric composition monitoring station of NDACC. The results cover the ozone and temperature measurements of 3 lidars permanently deployed at OHP (LTA, LiO3S and LiO3T), the mobile lidar STROZ from NASA-GSFC deployed at OHP for the occasion, co-located ECC ozonesondes, nearby radiosondes, and coincident satellite measurements from Aura-MLS and SABER. These inter-comparison campaigns are essential to characterize the performance of the ground-based instruments, often considered "reference measurements" when validating satellite-borne instruments, and when long-term intercalibration between these satellite measurements is needed. This, together with the intrinsic value of the OHP time series themselves, makes the publication of these results in AMT highly relevant. The take-home messages, as written in the abstract and conclusion, are clear and provide a good basis for reference in the future use of these datasets. Overall, the methods used are appropriate, but in several instances, a lack of clarity or rigor casts some doubts on the validity of some of the results, or more importantly, their interpretation. Two examples are 1) the loose/ambiguous reference to uncertainty and how it is used in the manuscript, and 2) the comparison of lidar and satellite uncertainty estimates and resulting conclusions. For this reason, major revisions are recommended before the manuscript can be considered for AMT publication. My comments and suggestions (major and minor points) are included below.

**We would like to take this time to thank Reviewer 1 for their keen insight into and frank evaluation of  our manuscript.  In particular, we believe that this article is much improved after a more careful use and discussion of terms associated with the different types of uncertainty involved in a measurement intercomparison campaign.  One key take away from this exercise is the need for a new publication which conducts a meta-analysis of NDACC validation campaigns and proposes new standardised language and procedures for future work.  In the companion article (set for submission in summer 2020) where we will present the lidar intercomparison and validation of the ozone and temperature lidar at the Hohenpeißenberg Meteorological Observatory, we shall endeavour to re-use the same language that has been suggested here.**

Major points:
1) Historically, the ozonesondes have typically been considered "independent" measurements. The ozone correction, as described here (i.e., using SAOZ), makes them dependent on the SAOZ measurement as well as the balloon blasting altitude. Recently, there has been a global effort for ozone sounding homogenization worldwide led by the SHADOZ community. Does this effort apply to the French ozonesonde program? Was the French ozonesonde team involved in

this effort? Please clarify the role of the ozonesondes: Are they considered reference or just correlative measurements?

**We have changed the direct translation of the French term 'facteur de correction (fc)' used in the article to 'quality control factor (qcf)' which is a better English equivalent phase.  This makes the text more clear that SAOZ is used to assure the quality of the ECC and not used to modify the data.**

**The ozonesondes are launched weekly as part of the NDACC France ozonesonde program by the full time technicians at OHP.  For this campaign the technicians were asked to prepare and launch a balloon every second night in July 2017 and nightly in March 2018.  The extra campaign launches should be consistent with the weekly ozonesonde record extending back 25 years (see: Gaudel et al. 2015).  Gérard Ancellet, the head of the French ozonesonde team and co-author on this paper, was responsible for processing the ozonesonde data.**

**Added text:**
**"The sondes and balloons were prepared and launched by the same OHP technicians responsible for the weekly ozonesonde launch.  The OHP radiosonde programme is homogenised under the auspices of NDACC France ozone measurements.  A new publication describing the full data treatment details, quality metrics, and uncertainty budget estimates is envisioned for 2021."**

**The ozonesondes and the satellites can be considered as correlative measurements as they are at times sampling different air masses than the lidars and are using different techniques.  We should properly interpret the lidar-lidar comparisons as strict 'reference measurements' and operate with an understanding that profile differences in lidar-balloon and lidar-satellite comparisons may have other sources, particularly at higher altitudes.**

2) It is difficult to figure out what the authors refer to as "uncertainty". For example, Page 17, line 9-10, "average measurement noise" is mentioned, and then in the same sentence "standard deviation of the ozone difference". Are the authors referring to the combined uncertainty of the two measurements? If so, please use "combined uncertainty" instead of "standard deviation".

**The sloppy writing has been addressed.  We have standardised 4 terms to address the different types of uncertainty: measurement uncertainty (uncertainty associated with the profiles of individual instruments), statistical uncertainty (variation between profiles from the same instrument), combined uncertainty (measurement + combined uncertainty), standard deviation (variation between profiles from different instruments).  Numerous modifications are made throughout the text.**

**Typo in referencing Equations 1 and 2 is fixed.**

**Added text:**
**"Given that for comparisons between any two pairs of lidar measurements during the LAVANDE campaign, there is nearly perfect spatio-temporal coincidence, we can neglect geophysical variations in our uncertainty budget. This is not true for lidar comparisons with sondes, satellites, or NCEP."**

3) Please make the clear distinction between what is random, what is systematic, and how these two types of uncertainty components are treated in the various parts of the manuscript. For example, in Page 20, line 16, it is claimed that the "uncertainty estimates. . .are too optimistic". Do these estimates account for systematic effects as well (total uncertainty?), or just the random component? If just random, it is not surprising that they do not match the r.m.s. differences, as r.m.s. will also reflect the presence of pseudo-systematic errors (e.g., alignment error for LiO3S, or aerosol interference for LTA).

**We have gone over the document and made explicit the type of uncertainty discussed in each case. See point 2. Measurement uncertainty profiles were supplied by each group as both r.m.s. and 'total measurement uncertainty'. For the blind intercomparison the 'total measurement uncertainty' was used.**

**Misalignment was not considered as a major contributor to the uncertainty budget as the alignment of each lidar was carefully checked and optimized before each measurement. Additionally, a significant misalignment would also impact the uncertainty comparisons for ozone and we do not see evidence for that.**

**The effect of aerosols in LTA temperatures is negligible above 30 km. In figure 19 we see that the standard deviation (grey) and combined uncertainty estimate (black) converge near 60 km.**

4) It is not clear what MLS single profile uncertainty is. Please clarify. A single profile uncertainty is used. Aren't several MLS profiles used in the comparisons? Please clarify.

**Added to P16L10:**
**"...associated with each individual 10 second profile. As was stated in \ref{sect:2.1.3} we use the same weighting technique on each of the associated measurement uncertainty profiles when calculating the 'nightly average' measurement uncertainty profile for collocated satellite overpasses."**

**Added to P6L20:**
**"The same three techniques were applied to the associated measurement uncertainty profiles to produce the nightly average measurement uncertainty profile (hereafter referred to simply as the 'measurement uncertainty'). In practice, these three versions of**

**the measurement uncertainty profiles were nearly identical showing that the statistical uncertainty on the measurement uncertainty is very low."**

5) The authors' interpretation of Fig 21 is overstated and inconsistent with that of the previous figure. The two sentences starting with "So this disagreement. . ." on page 34, line 8-9 assume that because disagreement is found for all lidars, then all lidars are "wrong". What if the source of the disagreement originates in SABER's underestimated uncertainty or a systematic error in the SABER profiles? If STROZ uncertainty is underestimated, why don't we see it in the comparison with MLS (Fig 20)?

**Yes, that is a very good point. It could very well be that the uncertainty is underestimated or a bias exists in SABER.  However, we cannot discount Figures 18 and 19 which clearly show that there is something that we are not accounting for which causes up to 2 K temperature differences between lidar measurements at low altitudes.**

**Figure 18 shows a 1.5 to 2 K difference below 30 km between the combined uncertainty of LiO3S and NASA vs the standard deviation of the two measurements (black vs grey curves).**
**Figure 19 shows up to 2 K difference below 50 km between the combined uncertainty of LTA and NASA vs the standard deviation of the two measurements (black vs grey curves).**
**Figure 20 shows that the standard deviation between NASA and MLS (grey) is in good agreement with the combined uncertainty (black). Further, the largest contribution to the combined uncertainty (black) comes from the MLS sampling uncertainty (light blue).**
**Figure 21 shows that the standard deviation between NASA and SABER (grey) is in poor agreement with the combined uncertainty (black) below 50 km. Sampling uncertainty for SABER (purple) is of the same order as the measurement uncertainty for SABER (pink) below 50 km.  At higher altitudes the sampling uncertainty (purple) is the largest contributor to the combined uncertainty (black).**

**We have replaced the two sentences:**
**"So this disagreement is fairly consistent across all the lidar instruments. All this indicates again that the very small temperature measurement uncertainty estimates of less than 1 K below 50 km for NASA-STROZ and other lidars are too optimistic. Additional uncertainty sources not considered in Leblanc et al. (2016c) may play a role (e.g. temporal changes in alignment,defocusing, multiple scattering etc.). From the LAVANDE results shown in Figs. 10 to 21 is seems that a combined temperature uncertainty of 1 to 3 K also below 50 km is not unrealistic for the participating lidar systems.**

**With a new statement:**

**"Given that in Figs. 18 and 19 we see a larger standard deviation between pairs of coincident lidar measurements (grey) than the estimated combined uncertainty (black) gives us reason to expect, we suggest that additional uncertainty sources not considered in Leblanc et al. (2016c) may play a role (e.g. temporal changes in alignment,defocusing, multiple scattering etc.). Additionally, the unexpectedly large standard deviation between the lidar and SABER results seen in Fig. 21 (grey), which may be due to unaccounted uncertainties in the SABER error budget, suggests a lower limit on the total temperature uncertainty budget of 1 to 3 K below 50 km. Taken together, these two suggestions imply that variations of approximately 3 K in the ensemble temperature differences seen in Fig. 15 is a reasonable threshold for validation of the participating lidar systems in the context of this LAVANDE campaign."**

6) Section 3.1.1: There is no evidence of vertical offset in figures 4 and 5. There is a difference in the shape of the peak between MLS and the others, but this does not seem to be the result of an altitude offset. For example, the MLS and GB ozone profiles are on top of each other at all altitudes above 24 km and all altitudes below 12 km. Also there is no evidence on these figures that SABER ozone/temp and MLS temp are shifted in altitude.

**Text removed:**
**"However, the satellite profiles of both ozone and temperature can be vertically offset from the profiles produced by the ground based instruments. In Fig. \ref{fig:o3_example_b} (left), we can see that the ozone maximum at 20 km and the sharp ozone decrease at 18 km, reported by the ground-based instruments, is shown 3 to 4 km lower in the MLS profile. This tendency for vertical offsets between lidar profiles and satellite profiles of temperature has been systematically documented over decades long time scales by \citet{wing2018b} and is attributed to systematic errors introduced in the retrieval of geopotential height is the satellite profiles.... MLS is in relatively poor agreement with all other instruments between 11 and 20 km with negative biases reaching $-40\%$, as shown in the left panel, while the ozonesondes and lidars compare much better in this region, with only 5$\%$ difference between them."**

**Figure 4 represents one of the best agreements for SABER. We can see a clear slope to the right in the SABER percent difference between approx 20 and 40 km. Taking the percent difference between two slightly shifted exponentials will give a linearly increasing percent difference like this. In this case we can see in the left hand side of Fig 4 that SABER between 26 and 36 km clearly has higher ozone than the other measurements and the percent difference on the right hand side is linearly sloped to the right going from approx -5% to 20%.**

**Here is a more extreme example of  SABER vs the other ozone measurements showing the vertical offset:**

[Figure]

Minor points:
The title should read "Observatoire de Haute Provence Âz˙

**Done**

Introduction: the authors should focus less on listing all the past campaigns and more on explaining the purpose of those campaigns and their outcome (include quantitative results as well as the main take-home messages from these campaigns.

**Our aim here is to give the larger context under which the LAVANDE campaign was conducted.  Given that other NDACC validation campaigns involved different instrument combinations, different locations and times of year, and different numbers of coincident measurements it is not clear how directly applicable these previous campaign results are with respect to LAVANDE.  The referee correctly points out our unsupported conclusion on P30L16 regarding direct comparisons of our results to those shown in Figure 10 of Leblanc et al. (that statement has been modified). We propose leaving the introductory text as is to avoid introducing new sources of confusion and complexity for the reader.**

**However, it would be a very good review article for someone to write a new meta-validation paper on NDACC intercomparisons.**

Page 3, Line 3: Add "Aura"

**Done**

Page 3, Line 29: Replace "off-line" by "non-absorbed"

**Done**

Page 4, 2.0.2.: The few technical details in this instrument description section do not convey the right message. Please specify that tropospheric DIAL requires more absorbing wavelengths (stronger UV) to measure ozone at ppb levels rather than ppm levels, which is why the wavelengths are different from stratospheric DIAL. Also, specify that the initial 266 nm beam is spectrally shifted by the Raman cell to produce 289 and 316 nm.

**Added that Raman emission is from the 266 nm source**

**Changed on and offline to absorbed and non-absorbed**

**Added "Using this Raman technique allows for the tropospheric lidar to measure much lower tropospheric ozone concentrations (on the order of ppb rather than ppm) as compared to the stratospheric system."**

Page 4, line 13: Remove "absolute"

**The lidar temperatures are absolute temperatures in Kelvin. We do not measure or report other types of temperature (potential temperature, brightness temperature, wet bulb temperature, temperature in Fahrenheit etc)**

Page 4, 2.0.3: As for paragraph 2.0.2., the choice of information included in this paragraph is somewhat arbitrary. There are other corrections applied to the signals to obtain the temperature profile (background noise, dead-time, molecular and particulate extinction). There is also a temperature initialization procedure at the top of the profile. I do not think the range-square correction should be mentioned without mentioning the other effects. I would recommend to add more details, or simply to mention that this is the backscatter temperature lidar technique, obtained by downward integration of atmospheric density (cf. Hauchecorne and Chanin).

**If we consider a lidar equation of the form:**

$$N(z) = C(z) * \left( \frac{n(z)}{z^2} \right) + B(z)$$

**Where all the corrections listed and more are combined into some complicated function C(z). Then we can imagine that for a perfect lidar system, operating in a perfectly linear regime, in a prefect atmosphere with a constant background the equation simplifies to:**

$$N(z) \approx \frac{n(z)}{z^2}$$

**The range scaled dependence is fundamental to the lidar equation in a way that other corrections like deadtime, overlap, particle extinctions are not. I can't imagine an idealized situation where this term disappears.**

**Added "algorithm details" to the last sentence of this paragraph**

Page 5, line 19: Is GPH converted to geometric altitude before it is used for comparison with lidar? Please specify.

**Added: "For comparison with the ground based lidars and ozonesondes the geopotential altitude is converted to a geometric altitude."**

Page 8, line 4: The impact of effective vertical resolution mentioned line 13 should be mentioned here

**We think that the text flows logically in its current form.**

Page 8, sentence starting with "The increased spring time variance": What is the purpose of this sentence? Is it supposed to introduce work further down in this manuscript? Please clarify.

**Yes. Figure 4 shows a summertime ozone profile and a springtime profile. We discuss the importance of conducting a 2 part validation campaign. One is dynamically active spring conditions (March 2018) and once in relatively more stable summer conditions (July 2017).**

Page 9, line 22: The large percent differences between MLS and the other instruments is unlikely to be associated with MLS vertical resolution. It is mainly because the ozone peak and ozone minimum are registered at different altitudes. MLS is capable of identifying these sharp transitions. The main reason for the observed difference is most likely the spatiotemporal coincidence and atmospheric variability

**This point seems inconsistent with point 6). Given that ozone and geopotential are measured separately it seems highly possible that there could be a distorted or shifted MLS ozone profile as a function of GPH.**

**Yes this is a fair point. Here is another example of a raw data plot where you can zoom in to see the features and resolution of the sonde and tropospheric lidar which are not reproduced by MLS. The temporal offset should be very low as on this particular night 20170712 the lidars measured until 2 am and we incorporated two MLS overpasses -- one to the west (approx 870 km) at local midnight and one to the west at approximately 1:40 am (approx. 950 km). However, the passage of a high pressure system around local midnight could mean that we are measuring completely different air masses in the**

**troposphere.**

[Figure]

We have replaced line 22 with:
"... where the low vertical resolution of MLS cannot resolve the fine layers of the dynamic lower stratosphere." with "... where differences in spatiotemporal coincidence and atmospheric variability can lead to the sampling of different air masses."

Page 10, line 3-4: This is inconsistent with the figure. In fact, the best agreement is below 25 km

**Clarified by replacing:**
"In general, SABER ozone does not agree with ozone measurements from the other instruments below 25 km as it is principally an instrument focused on the upper middle atmosphere; hence it is not plotted for this altitude range. The extent of the disagreement can be an order of magnitude larger than the differences between the ozone concentration measured by the other instruments."

**With:**
**"In most cases, SABER ozone does not agree with ozone measurements from the other instruments below 25 km as it is principally an instrument focused on the upper middle atmosphere.  The extent of the disagreement can be an order of magnitude larger than the differences between the ozone concentration measured by the other instruments. We will revisit this topic later in the article when discussing the ensemble ozone differences in Fig. 7."**

Page 10, line 8: I do not see any disagreement in the altitude. The peak is just smoother, and SABER actually reproduces well the ozone minimum right below the peak

**Sentence has been removed.**

Page 11, line3: I do not think the addition of a new Raman channel will reduce the warm bias. Please rephrase

**Adding a new rotational Raman channel would allow for better measurements of tropospheric temperature.**

Page 13, line 5: Typo

**I don't see it:**
**The heavier smoothing and integration is required above 40 km due to the drop in the lidar signal to noise ratio.**

Page 17, line 3: Please clarify. Does "uncertainty estimated by the retrievals" include only random components (photon noise), or is it the total uncertainty? How is this average computed?

**Text has been changed to address concerns raised in point 2)**

**A simple average is calculated.**

Page 17, Equation 1: Define L and N

**Done**

Page 20, line 8-12: Please define "MLS individual profile uncertainty"? Is that precision (random) or total uncertainty? Shouldn't a "campaign mean" of the individual uncertainties be considered instead of a single profile uncertainty? (just like it was done for GB instruments)

**Text has been changed to address concerns raised in point 2)**

Correlation diagnostics (section 4, page 15 and section 5): What is the purpose of the correlation diagnostic? This diagnostic seems to introduce more confusion than clarification on the origins of the differences between the instruments. For example the authors state that the method is sensitive to the size of the averaging window, "drastically increasing or decreasing the amplitude of this peak". Please clarify or remove this part to keep the discussions of Figures 7 and 15.

**The key point arises on P15L9-10 where we explain that "When the co-variance of the data, arising from real differences in ozone concentration drops faster than the variance of the data, in part arising from statistical scatter, we see a resulting drop in the correlation."  I think it adds another layer of understanding to the development of our understanding of the data:**
**Fig. 4 is a single example with percent difference (illustrates each type of measurement);**
**Fig. 6 shows the temporal evolution of the measurements (including the difference in the geophysical variability between summer 2018 and spring 2018);**
**Fig. 7 tells us what the average percent difference is (can be difficult to interpret the significance at high vs low ozone concentrations);**
**Fig 8 (left). Shows that the large percent differences in Fig. 7 represent real differences in the measurement (particularly at low ozone concentration for MLS and SABER where the ratio can be 5 or 6 to 1)**
**Fig.8.(centre) shows very tight data clustering in the region sound the ozone max**
**Fig. 8(right) also shows very tight data clustering along the 1:1 black reference line (we can use this to modify our understanding of the large percent differences above 40 km in Fig. 7.  They do not correspond to very large changes in O3 number density**
**Fig. 9 Shows the extent to which observed variance in the data can be explained by the co-variance between the data as a profiles with altitude.**

 Page 25, line 1: What is the a priori source? What is the altitude of initialization? Is it the same altitude for LTA and STROZ? LiO3S initialization is much lower. What a priori do they use? In order to investigate the STROZ warm bias at the top, could an alternate data processing be done using the same a priori at the same altitude? This would remove any bias associated with the tie-on procedure.

**These are very good points.  We discussed the possibility of including this information and intercomparison in the LAVANDE article.  It was decided that this paper was already very long and if we start discussing and changing a priori assumptions for temperature initialization then the study would no longer be 'blind'.**

**We have added the following: "A full study of the effects of the a priori selection, initialization altitude, and tie-on uncertainty would be a good topic for another NDACC algorithm validation article where we are not constrained by the need to perform a 'blind' comparison."**

Page 27, line 8, "between 60 and 80 km": This sentence is misleading. If STROZ and LTA use the same a priori source (MSIS?), it is not surprising that the correlation increases as the profiles approach their tie-on altitude. This high correlation does not demonstrate instrument performance.

**For the purposes of this blind intercomparison I'm not going to add additional text. We would like this comparison to be based solely on the information provided to the NDACC campaign referee (Wolfgang Steinbrecht) by the participating instrument PIs.**

**I'm not sure that the tie-on error is very significant at 60 km. Good practice in the Hauchecorne-Chanin method is to cut the top 2 scale heights off from the final temperature profile. So I would imagine that initialization in both systems happens above 90 km.**

Page 30, line 7: "beyond statistical uncertainty": Systematic uncertainty components must be included in Figure 18, especially if they are not negligible, for example, uncertainty associated with temperature initialization and background noise correction in the mesosphere, and possibly dead-time correction uncertainty at the bottom of the profiles

**Agreed. We noted that future work is planned to refine the temperature error budgets.**

Page 30, line 16, ". . .and also the temperature estimate of fig 10 of Leblanc et al.) are too optimistic..": I do not understand this sentence. Leblanc's figure 10 shows an example of uncertainty budget for a different lidar system (unrelated to LAVANDE), not including the impact of aerosols or misalignment. It is not surprising to find different results here, especially if the LTA profiles are impacted by aerosol and/or misalignment.

**Removed:**
**"(and also the temperature estimate in Fig. 10 of Leblanc et al., 2016c)"**

Section 5: There is no attempt to explain the NCEP differences (SSU?, AMSU?) Any published reference? Did the authors consider using MERRA-2?

**If I remember correctly there isn't much data from AMSU above 40 km (I think mostly channel 14 maybe channel 13 as well). Differences above these attitudes should not be surprising.**

**Added to P26L4:**
**"...which may in part be due to the vertical averaging and data density differences between lidar measurements and AMSU as demon-strated by Funatsu et al. (2008)"**

**Yes, MERRA2 or the new ERA5 may be good to use in the future.**

Page 37, line 3, "Other sources of uncertainty": The authors are correct that other sources of uncertainty must probably be accounted for. But they should also discuss the possibility of optimizing the instrument set up (in this case alignment) so that errors are minimized and the introduction of additional uncertainty sources is less relevant.

**Added to P37L3:**
**"... or that further work can be done in addressing potential sources of measurement bias (e.g. alignment, a priori temperature initialization, deadtime corrections)"**

Figures: Figures 2-3: are too small.

**Changed to 70% text width**

Figure 4: On the right panel, the differences between the instruments are not shown below 8-10 km. Please plot the differences It would be good to uniformize the instrument short names throughout all figures and text. For example, sometimes, we see "OHP 532 nm", sometimes "LTA".

**The percent differences are with respect to NASA-STROZ which ends around 10 km.**

---

## Author Comment (AC2) · 8 Jul 2020

**Response R2: "Intercomparison and Evaluation of Ground- and Satellite-Based Stratospheric Ozone and Temperature profiles above Observatoire Haute Provence during the Lidar Validation NDACC Experiment (LAVANDE)"**

The study presents a detailed comparison of the stationary lidars at OHP with a series of other data sets, in particular a mobile lidar system from NASA. Also ECC, MLS, SABER ozone and temperature profiles are considered. The comparison has been performed "blind" by an impartial expert, which is a very interesting approach. For the comparison, a wide range of visualizations is presented reaching from mean profiles, over time series and scatter plots to correlation profiles. This gives a detailed insight in the behavior of each data set but makes the paper a bit lengthy. The study also evaluates in detail the uncertainties of each data set, which is a very interesting aspect and should be done more in validation studies. Validations studies such as the one presented here are important contributions to understand instrumental differences and to obtain consistent long term data sets. Methods and results are well explained and the paper follows a logic structure. I recommend the paper for publication in AMT and provide below minor comments for the authors to consider.

**Thank you for your comments regarding the intercomparison of satellite and ground based lidar measurements. This is a very important (and difficult) problem for all of us. It is our hope that NDACC data, in particular our long data set of lidar temperatures and ozone measurements can be presented in a way that is useful for long-term comparisons with space based instruments. If we can develop robust procedures for determining coincidence with satellites perhaps near-real time comparisons can be made and automatically uploaded to the NDACC website.**

Minor comments In the effort to identify co-located profiles between lidar and satellite the authors allow a time difference of up to 12 hours. Given the diurnal cycles in temperature and ozone in the stratosphere and mesosphere, this seems too tolerant. Have any effects related to tides and diurnal cycles been corrected? An analysis of the distribution of the time differences would be helpful to convince the reader that systematic biases are not a consequence of diurnal cycles.

**The 12 hour window mostly applies to SABER measurements. It is difficult to achieve very close spatio-temporal matching and have a sufficient number of coincident measurements to have a statistically meaningful comparison with a non-sunsynchronous satellite. We only have 28 nights of lidar measurements with which to conduct the intercomparison exercise. Wing et al. 2018 discusses the pros and cons of this trade off in more detail. Another study in JGR:Atmospheres by Dawkins et al. 2018 "Validation of SABER v2.0 operational temperature data with ground‐based lidars in the mesosphere‐lower thermosphere region (75–105 km)" Also has a very good discussion**

**on this topic (however given the upper mesospheric and lower thermospheric focus it is not directly applicable to the work in LAVANDE)**

**For MLS there are generally one or two overpasses included in the nightly coincidence criteria. Generally around 1:40 am local time plus or minus approximately 99 minutes. With the sun synchronous satellite the geographic constraint is more strict than the temporal constraint.**

The weighted mean on p6, l16 takes into account the typical wind speed in the stratosphere. What is the justification for this? Diurnal cycles in temperature and ozone are not driven by advection but by photochemistry and tidal waves. Please comment.

**Our objective here is to try and weight profiles by both distance and time. From a simple mathematical standpoint we need to assume a wind speed to make the units agree. From a geophysical standpoint, we can have cases where there are two or more satellite overpasses in our geographic coincidence box at different times. For example, we could have a lidar measurement centered around local midnight with a satellite overpass 1000 km to the west and a second satellite overpass 100 minutes later 700 km to the east. However, during the period between the two satellite overpasses the air is generally advected from west to east over the lidar site. (See our reply to R1 with respect to tropospheric variability and local frontal systems). The use of an assumed stratospheric wind speed is an attempt to correct for the relative motion of the atmosphere with respect to the lidar station over these short timescales.**

**We recognise that this is not a perfect correction but given our recent experiences with validating Aeolus wind measurements using the Doppler wind lidar at OHP approximately 10 m/s is not an unreasonable assumption for late spring and summer. Khaykin, Sergey M., et al. "Doppler lidar at Observatoire de Haute-Provence for wind profiling up to 75 km altitude: performance evaluation and observations."** *Atmospheric Measurement Techniques* **13.3 (2020).**

P6, l11: do you mean 10 to 20 matching profiles per night? With the chosen wording this is not absolutely clear

**Replaced with:**
**"It results in between 10 to 20 coincident profiles for MLS and SABER, which are generally divided between one or two satellite overpasses, for a given night during the LAVANDE campaign."**